# Algorithm-Dependent Generalization Bounds for Overparameterized Deep Residual Networks

**Spencer Frei**[*] and **Yuan Cao**[†] and **Quanquan Gu**[‡]

## Abstract

The skip-connections used in residual networks have become a standard architecture choice in deep learning due to the increased training stability and generalization performance with this architecture, although there has been limited theoretical understanding for this improvement. In this work, we analyze overparameterized deep residual networks trained by gradient descent following random initialization, and demonstrate that (i) the class of networks learned by gradient descent constitutes a small subset of the entire neural network function class, and (ii) this subclass of networks is sufficiently large to guarantee small training error. By showing (i) we are able to demonstrate that deep residual networks trained with gradient descent have a small generalization gap between training and test error, and together with (ii) this guarantees that the test error will be small. Our optimization and generalization guarantees require overparameterization that is only logarithmic in the depth of the network, while all known generalization bounds for deep non-residual networks have overparameterization requirements that are at least polynomial in the depth. This provides an explanation for why residual networks are preferable to non-residual ones.

## 1  Introduction

Deep learning has seen an incredible amount of success in a variety of settings over the past eight years, from image recognition [15] to audio recognition [20] and more. Compared with its rapid and widespread adoption, the theoretical understanding of why deep learning works so well has lagged significantly. This is particularly the case in the common setup of an overparameterized network, where the number of parameters in the network greatly exceeds the number of training examples and input dimension. In this setting, networks have the capacity to perfectly fit training data, regardless of if it is labeled with real labels or random ones [25]. However, when trained on real data, these networks also have the capacity to truly learn patterns in the data, as evidenced by the impressive performance of overparameterized networks on a variety of benchmark datasets. This suggests the presence of certain mechanisms underlying the data, neural network architectures, and training algorithms which enable the generalization performance of neural networks. A theoretical analysis that seeks to explain why neural networks work so well would therefore benefit from careful attention to the specific properties that neural networks have when trained under common optimization techniques.

Many recent attempts at uncovering the generalization ability of deep learning focused on general properties of neural network function classes with fixed weights and training losses. For instance,

---

[*]Department of Statistics, University of California, Los Angeles, CA 90095, USA; e-mail: `spencerfrei@ucla.edu`

[†]Department of Computer Science, University of California, Los Angeles, CA 90095, USA; e-mail: `yuancao@cs.ucla.edu`

[‡]Department of Computer Science, University of California, Los Angeles, CA 90095, USA; e-mail: `qgu@cs.ucla.edu`

Bartlett et al. [4] proved spectrally normalized margin bound for deep fully connected networks in terms of the spectral norms of the weights at each layer. Neyshabur et al. [18] proved a similar bound using PAC-Bayesian approach. Arora et al. [2] developed a compression-based framework for generalization of deep fully connected and convolutional networks, and also provided an explicit comparison of recent generalization bounds in the literature. All these studies involved algorithm-independent analyses of the neural network generalization, with resultant generalization bounds that involve quantities that make the bound looser with increased overparameterization.

An important recent development in the practical deployment of neural networks has been the introduction of skip connections between layers, leading to a class of architectures known as residual networks. Residual networks were first introduced by He et al. [13] to much fanfare, quickly becoming a standard architecture choice for state-of-the-art neural network classifiers. The motivation for residual networks came from the poor behavior of very deep traditional fully connected networks: although deeper fully connected networks can clearly express any function that a shallower one can, in practice (i.e. using gradient descent) it can be difficult to choose hyperparameters that result in small training error. Deep residual networks, on the other hand, are remarkably stable in practice, in the sense that they avoid getting stuck at initialization or having unpredictable oscillations in training and validation error, two common occurrences when training deep non-residual networks. Moreover, deep residual networks have been shown to generalize with better performance and far fewer parameters than non-residual networks [22, 7, 14]. We note that much of the recent neural network generalization literature has focused on non-residual architectures [4, 18, 2, 12, 5] with bounds for the generalization gap that grow exponentially as the depth of the network increases. Li et al. [16] recently studied a class of residual networks and proved algorithm-independent bounds for the generalization gap that become larger as the depth of the network increases, with a dependence on the depth that is somewhere between sublinear and exponential (a precise characterization requires further assumptions and/or analysis). We note that verifying the non-vacuousness of algorithm-independent generalization bounds relies on empirical arguments about what values the quantities that appear in the bounds generally take in practical networks (i.e. norms of weight matrices and interlayer activations), while algorithm-dependent generalization bounds such as the ones we provide in this paper can be understood without relying on experiments.

## 1.1 Our Contributions

In this work, we consider fully connected deep ReLU residual networks and study optimization and generalization properties of such networks that are trained with discrete time gradient descent following Gaussian initialization.

We consider binary classification under the cross-entropy loss and focus on data that come from distributions $\mathcal{D}$ for which there exists a function $f$ for which $y \cdot f(x) \geq \gamma > 0$ for all $(x, y) \in \operatorname{supp} \mathcal{D}$ from a large function class $\mathcal{F}$ (see Assumption 3.2). By analyzing the trajectory of the parameters of the network during gradient descent, for any error threshold $\varepsilon > 0$, we are able to show:

1. Under the cross-entropy loss, we can study an analogous surrogate error and bound the true classification error by the true surrogate error. This method was introduced by Cao and Gu [5].

2. If $m^* = \tilde{O}(\operatorname{poly}(\gamma^{-1})) \cdot \max(d, \varepsilon^{-2})$, then provided every layer of the network has at least $m \geq m^*$ units, gradient descent with small enough step size finds a point with empirical surrogate error at most $\varepsilon$ in at most $\tilde{O}(\operatorname{poly}(\gamma^{-1}) \cdot \varepsilon^{-1})$ steps with high probability. Here, $\tilde{O}(\cdot)$ hides logarithmic factors that may depend on the depth $L$ of the network, the margin $\gamma$, number of samples $n$, error threshold $\varepsilon$, and probability level $\delta$.

3. Provided $m^* = \tilde{O}(\operatorname{poly}(\gamma^{-1}, \varepsilon^{-1}))$ and $n = \tilde{O}(\operatorname{poly}(\gamma^{-1}, \varepsilon^{-1}))$, the difference between the empirical surrogate error and the true surrogate error is at most $\varepsilon$ with high probability, and therefore the above provide a bound on the true classification error of the learned network.

We emphasize that our guarantees above come with at most logarithmic dependence on the depth of the network. Our methods are adapted from those used in the fully connected architecture by Cao and Gu [5] to the residual network architecture. The main proof idea is that overparameterization forces gradient descent-trained networks to stay in a small neighborhood of initialization where the learned

networks (i) are guaranteed to find small surrogate training error, and (ii) come from a sufficiently small hypothesis class to guarantee a small generalization gap between the training and test errors. By showing that these competing phenomena occur simultaneously, we are able to derive the test error guarantees of Corollary 3.7. The key insight of our analysis is that the Lipschitz constant of the network output for deep residual networks as well as the semismoothness property (Lemma 4.2) have at most logarithmic dependence on the depth, while the known analogues for non-residual architectures all have polynomial dependence on the depth.

## 1.2 Additional Related Work

In the last year there has been a variety of works developing algorithm-dependent guarantees for neural network optimization and generalization [17, 1, 28, 9, 3, 5, 27, 6]. Li and Liang [17] were among the first to theoretically analyze the properties of overparameterized fully connected neural networks trained with Gaussian random initialization, focusing on a two layer (one hidden layer) model under a data separability assumption. Their work provided two significant insights into the training process of overparameterized ReLU neural networks: (1) the weights stay close to their initial values throughout the optimization trajectory, and (2) the ReLU activation patterns for a given example do not change much throughout the optimization trajectory. These insights were the backbone of the authors' strong generalization result for stochastic gradient descent (SGD) in the two layer case. The insights of Li and Liang [17] provided a basis to various subsequent studies. Du et al. [9] analyzed a two layer model using a method based on the Gram matrix using inspiration from kernel methods, showing that gradient descent following Gaussian initialization finds zero training loss solutions at a linear rate. Zou et al. [28] and Allen-Zhu et al. [1] extended the results of Li and Liang to the arbitrary $L$ hidden layer fully connected case, again considering (stochastic) gradient descent trained from random initialization. Both authors showed that, provided the networks were sufficiently wide, arbitrarily deep networks would converge to a zero training loss solution at a linear rate, using an assumption about separability of the data. Recently, Zou and Gu [27] provided an improved analysis of the global convergence of gradient descent and SGD for training deep neural networks, which enjoys a milder over-parameterization condition and better iteration complexity than previous work. Under the same data separability assumption, Zhang et al. [26] showed that deep residual networks can achieve zero training loss for the squared loss at a linear rate with overparameterization essentially independent of the depth of the network. We note that Zhang et al. [26] studied optimization for the regression problem rather than classification, and their results do not distinguish the case with random labels from that with true labels; hence, it is not immediately clear how to translate their analysis to a generalization bound for classification under the cross-entropy loss as we are able to do in this paper.

The above results provide a concrete answer to the question of why overparameterized deep neural networks can achieve zero training loss using gradient descent. However, the theoretical tools of Du et al. [9], Allen-Zhu et al. [1], Zou et al. [28], Zou and Gu [27] apply to data with random labels as well as true labels, and thus do not explain the generalization to unseen data observed experimentally. Dziugaite and Roy [10] optimized PAC-Bayes bounds for the generalization error of a class of stochastic neural networks that are perturbations of standard neural networks trained by SGD. Cao and Gu [5] proved a guarantee for arbitrarily small generalization error for classification in deep fully connected neural networks trained with gradient descent using random initialization. The same authors recently provided an improved result for deep fully connected networks trained by stochastic gradient descent using a different approach that relied on the neural tangent kernel and online-to-batch conversion [6]. E et al. [11] recently developed algorithm-dependent generalization bounds for a special residual network architecture with many different kinds of skip connections by using kernel methods.

## 2 Network Architecture and Optimization Problem

We begin with the notation of the paper. We denote vectors by lowercase letters and matrices by uppercase letters, with the assumption that a vector $v$ is a column vector and its transpose $v^\top$ is a row vector. We use the standard $O(\cdot), \Omega(\cdot), \Theta(\cdot)$ complexity notations to ignore universal constants, with $\tilde{O}(\cdot), \tilde{\Omega}(\cdot)$ additionally ignoring logarithmic factors. For $n \in \mathbb{N}$, we write $[n] = \{1, 2, \ldots, n\}$. Denote the number of hidden units at layer $l$ as $m_l$, $l = 1, \ldots, L + 1$. Let the $l$-th layer weights be $W_l \in \mathbb{R}^{m_{l-1} \times m_l}$, and concatenate all of the layer weights into a vector $W = (W_1, \ldots, W_{L+1})$.

Denote by $w_{l,j}$ the $j$-th column of $W_l$. Let $\sigma(x) = \max(0, x)$ be the ReLU nonlinearity, and let $\theta$ be a constant scaling parameter. We consider a class of residual networks defined by the following architecture:

$$x_1 = \sigma(W_1^\top x), \qquad x_l = x_{l-1} + \theta\sigma\left(W_l^\top x_{l-1}\right), \ \ l = 2, \ldots, L,$$
$$x_{L+1} = \sigma(W_{L+1}^\top x_L).$$

Above, we denote $x_l$ as the $l$-th hidden layer activations of input $x \in \mathbb{R}^d$, with $x_0 := x$. In order for this network to be defined, it is necessary that $m_1 = m_2 = \cdots = m_L$. We are free to choose $m_{L+1}$, as long as $m_{L+1} = \Theta(m_1)$ (see Assumption 3.4). We define a constant, non-trainable vector $v = (1, 1, \ldots, 1, -1, -1, \ldots, -1)^\top \in \mathbb{R}^{m_{L+1}}$ with equal parts $+1$ and $-1$'s that determines the network output,

$$f_W(x) = v^\top x_{L+1}.$$

We note that our methods can be extended to the case of a trainable top layer weights $v$ by choosing the appropriate scale of initialization for $v$. We choose to fix the top layer weights in this paper for simplicity of exposition.

We will find it useful to consider the matrix multiplication form of the ReLU activations, which we describe below. Let $\mathbb{1}(A)$ denote the indicator function of a set $A$, and define diagonal matrices $\Sigma_l(x) \in \mathbb{R}^{m_l \times m_l}$ by $[\Sigma_l(x)]_{j,j} = \mathbb{1}(w_{l,j}^\top x_{l-1} > 0)$, $l = 1, \ldots, L+1$. By convention we denote products of matrices $\prod_{i=a}^{b} M_i$ by $M_b \cdot M_{b-1} \cdot \ldots \cdot M_a$ when $a \leq b$, and by the identity matrix when $a > b$. With this convention, we can introduce notation for the $l$-to-$l'$ interlayer activations $H_l^{l'}(x)$ of the network. For $2 \leq l \leq l' \leq L$ and input $x \in \mathbb{R}^d$ we denote

$$H_l^{l'}(x) := \prod_{r=l}^{l'} \left(I + \theta\Sigma_r(x)W_r^\top\right). \qquad\qquad (2 \leq l \leq l' \leq L) \qquad\qquad (1)$$

If $l = 1 < l'$, we denote $H_1^{l'}(x) = H_2^{l'}(x)\Sigma_1(x)W_1^\top$, and if $l' = L+1 > l$, we denote $H_l^{L+1}(x) = \Sigma_{L+1}(x)W_{L+1}^\top H_l^L(x)$. Using this notation, we can write the output of the neural network as $f_W(x) = v^\top H_{l+1}^{L+1}(x)x_l$ for any $l \in \{0\} \cup [L+1]$ and $x \in \mathbb{R}^d$. For notational simplicity, we will denote $\Sigma_l(x)$ by $\Sigma_l$ and $H_l^{l'}(x)$ by $H_l^{l'}$ when the dependence on the input is clear.

We assume we have i.i.d. samples $(x_i, y_i)_{i=1}^n \sim \mathcal{D}$ from a distribution $\mathcal{D}$, where $x_i \in \mathbb{R}^d$ and $y_i \in \{\pm 1\}$. We note the abuse of notation in the above, where $x_l \in \mathbb{R}^{m_l}$ refers to the $l$-th hidden layer activations of an arbitrary input $x \in \mathbb{R}^d$ while $x_i$ refers to the $i$-th sample $x_i \in \mathbb{R}^d$. We shall use $x_{l,i} \in \mathbb{R}^{m_l}$ when referring to the $l$-th hidden layer activations of a sample $x_i \in \mathbb{R}^d$ (where $i \in [n]$ and $l \in [L+1]$), while $x_l \in \mathbb{R}^{m_l}$ shall refer to the $l$-th hidden layer activation of arbitrary input $x \in \mathbb{R}^d$.

Let $\ell(x) = \log(1 + \exp(-x))$ be the cross-entropy loss. We consider the empirical risk minimization problem optimized by constant step size gradient descent,

$$\min_W L_S(W) := \frac{1}{n}\sum_{i=1}^n \ell(y_i \cdot f_W(x_i)), \qquad W_l^{(k+1)} = W_l^{(k)} - \eta \cdot \nabla_{W_l} L_S(W^{(k)}) \quad (l \in [L+1]).$$

We shall see below that a key quantity for studying the trajectory of the weights in the above optimization regime is a surrogate loss defined by the derivative of the cross-entropy loss. We denote the empirical and true surrogate loss by

$$\mathcal{E}_S(W) := -\frac{1}{n}\sum_{i=1}^n \ell'(y_i \cdot f_W(x_i)), \quad \mathcal{E}_\mathcal{D}(W) := \mathbb{E}_{(x,y)\sim\mathcal{D}}[-\ell'(y \cdot f_W(x))],$$

respectively. The empirical surrogate loss was first introduced by Cao and Gu [5] for the study of deep non-residual networks. Finally, we note here a formula for the gradient of the output of the network with respect to different layer weights:

$$\nabla_{W_l} f_W(x) = \theta^{\mathbb{1}(2 \leq l \leq L)} x_{l-1} v^\top H_{l+1}^{L+1} \Sigma_l(x), \qquad\qquad (1 \leq l \leq L+1). \qquad\qquad (2)$$

# 3 Main Theory

We first go over the assumptions necessary for our proof and then shall discuss our main results. Our assumptions align with those made by Cao and Gu [5] in the fully connected case. The first main assumption is that the input data is normalized.

**Assumption 3.1.** Input data are normalized: $\text{supp}(\mathcal{D}_x) \subset S^{d-1} = \{x \in \mathbb{R}^d : \|x\|_2 = 1\}$.

Data normalization is common in statistical learning theory literature, from linear models up to and including recent work in neural networks [17, 28, 9, 1, 3, 5], and can easily be satisfied for arbitrary training data by mapping samples $x \mapsto x/\|x\|_2$.

The next assumption is on the data generating distribution. Because overparameterized networks can memorize data, any hope of demonstrating that neural networks have a small generalization gap must restrict the class of data distribution processes to one where some type of learning is possible.

**Assumption 3.2.** Let $p(u)$ denote the density of a standard $d$-dimensional Gaussian vector. Define

$$\mathcal{F} = \left\{ \int_{\mathbb{R}^d} c(u)\sigma(u^\top x)p(u)\mathrm{d}u \, : \, \|c(\cdot)\|_\infty \leq 1 \right\}.$$

Assume there exists $f(\cdot) \in \mathcal{F}$ and constant $\gamma > 0$ such that $y \cdot f(x) \geq \gamma$ for all $(x, y) \in \text{supp}(\mathcal{D})$.

Assumption 3.2 was introduced by Cao and Gu [5] for the analysis of fully connected networks and is applicable for distributions where samples can be perfectly classified by the random kitchen sinks model of Rahimi and Recht [19]. One can view a function from this class as the infinite width limit of a one-hidden-layer neural network with regularizer given by a function $c(\cdot)$ with bounded $\ell^\infty$-norm. As pointed out by Cao and Gu [5], this assumption includes the linearly separable case.

Our next assumption concerns the scaling of the weights at initialization.

**Assumption 3.3** (Gaussian initialization). We say that the weight matrices $W_l \in \mathbb{R}^{m_{l-1} \times m_l}$ are generated via Gaussian initialization if each of the entries of $W_l$ are generated independently from $N(0, 2/m_l)$.

This assumption is common to much of the recent theoretical analyses of neural networks [17, 28, 1, 9, 3, 5] and is known as the He initialization due to its usage in the first ResNet paper by He et al. [13]. This assumption guarantees that the spectral norms of the weights are controlled at initialization.

Our last assumption concerns the widths of the networks we consider and allows us to exclude pathological dependencies between the width and other parameters that define the architecture and optimization problem.

**Assumption 3.4** (Widths are of the same order). We assume $m_{L+1} = \Theta(m_L)$. We call $m = m_L \wedge m_{L+1}$ the width of the network.

Our first theorem shows that provided we have sufficient overparameterization and sufficiently small step size, the iterates $W^{(k)}$ of gradient descent stay within a small neighborhood of their initialization. Additionally, the empirical surrogate error can be bounded by a term that decreases as we increase the width $m$ of the network.

**Theorem 3.5.** Suppose $W^{(0)}$ are generated via Gaussian initialization and that the residual scaling parameter satisfies $\theta = 1/\Omega(L)$. For $\tau > 0$, denote a $\tau$-neighborhood of the weights $W^{(0)} = (W_1^{(0)}, \ldots, W_{L+1}^{(0)})$ at initialization by

$$\mathcal{W}(W^{(0)}, \tau) := \left\{ W = (W_1, \ldots, W_{L+1}) : \left\| W_l - W_l^{(0)} \right\|_F \leq \tau \; \forall l \in [L+1] \right\}.$$

There exist absolute constants $\nu, \nu', \nu'', C, C' > 0$ such that for any $\delta > 0$, provided $\tau \leq \nu\gamma^{12} (\log m)^{-\frac{3}{2}}$, $\eta \leq \nu'(\tau m^{-\frac{1}{2}} \wedge \gamma^4 m^{-1})$, and $K\eta \leq \nu''\tau^2\gamma^4 (\log(n/\delta))^{-\frac{1}{2}}$, then if the width of the network is such that,

$$m \geq C' \left( \tau^{-\frac{4}{3}} d \log \frac{m}{\tau\delta} \vee d \log \frac{mL}{\delta} \vee \tau^{-\frac{2}{3}} (\log m)^{-1} \log \frac{L}{\delta} \vee \gamma^{-2} \left( d \log \frac{1}{\gamma} \vee \log \frac{L}{\delta} \right) \vee \log \frac{n}{\delta} \right)$$

then with probability at least $1 - \delta$, gradient descent starting at $W^{(0)}$ with step size $\eta$ generates $K$ iterates $W^{(1)}, \ldots, W^{(K)}$ that satisfy:

(i) $W^{(k)} \in \mathcal{W}(W^{(0)}, \tau)$ for all $k \in [K]$.

(ii) There exists $k \in \{0, \ldots, K-1\}$ with $\mathcal{E}_S(W^{(k)}) \leq C \cdot m^{-\frac{1}{2}} \cdot (K\eta)^{-\frac{1}{2}} \left(\log \frac{n}{\delta}\right)^{\frac{1}{4}} \cdot \gamma^{-2}$.

This theorem allows us to restrict our attention from the large class of all deep residual neural networks to the reduced complexity class of those with weights that satisfy $W \in \mathcal{W}(W^{(0)}, \tau)$. Our analysis provides a characterization of the radius of this reduced complexity class in terms of parameters that define the network architecture and optimization problem. Additionally, this theorem allows us to translate the optimization problem over the empirical loss $L_S(W)$ into one for the empirical surrogate loss $\mathcal{E}_S(W^{(k)})$, a quantity that is simply related to the classification error (its expectation is bounded by a constant multiple of the classification error under 0-1 loss; see Appendix A.2).

Our next theorem characterizes the Rademacher complexity of the class of residual networks with weights in a $\tau$-neighborhood of the initialization. Additionally, it connects the test accuracy with the empirical surrogate loss and the Rademacher complexity.

**Theorem 3.6.** Let $W^{(0)}$ denote the weights at Gaussian initialization and suppose the residual scaling parameter satisfies $\theta = 1/\Omega(L)$. Suppose $\tau \leq 1$. Then there exist absolute constants $C_1, C_2, C_3 > 0$ such that for any $\delta > 0$, provided

$$m \geq C_1 \left( \tau^{-\frac{2}{3}} (\log m)^{-1} \log(L/\delta) \vee \tau^{-\frac{4}{3}} d \log(m/(\tau\delta)) \vee d \log(mL/\delta) \right),$$

then with probability at least $1 - \delta$, we have the following bound on the Rademacher complexity,

$$\mathfrak{R}_n \left( \{ f_W : W \in \mathcal{W}(W^{(0)}, \tau) \} \right) \leq C_2 \left( \tau^{\frac{4}{3}} \sqrt{m \log m} + \frac{\tau\sqrt{m}}{\sqrt{n}} \right),$$

so that for all $W \in \mathcal{W}(W^{(0)}, \tau)$,

$$\mathbb{P}_{(x,y)\sim\mathcal{D}} \left( y \cdot f_W(x) < 0 \right) \leq 2\mathcal{E}_S(W) + C_2 \left( \tau^{\frac{4}{3}} \sqrt{m \log m} + \frac{\tau\sqrt{m}}{\sqrt{n}} \right) + C_3 \sqrt{\frac{\log(1/\delta)}{n}}. \quad (3)$$

We shall see in Section 4 that we are able to derive the above bound on the Rademacher complexity by using a semi-smoothness property of the neural network output and an upper bound on the gradient of the network output. Standard arguments from statistical learning theory provide the first and third terms in (3).

The missing ingredients needed to realize the result of Theorem 3.6 for networks trained by gradient descent are supplied by Theorem 3.5, which gives (i) control of the growth of the empirical surrogate error $\mathcal{E}_S$ along the gradient descent trajectory, and (ii) the distance $\tau$ from initialization before which we are guaranteed to find small empirical surrogate error. Putting these together yields Corollary 3.7.

**Corollary 3.7.** Suppose that the residual scaling parameter satisfies $\theta = 1/\Omega(L)$. Let $\varepsilon, \delta > 0$ be fixed. Suppose that $m^* = \tilde{O}(\text{poly}(\gamma^{-1})) \cdot \max(d, \varepsilon^{-14}) \cdot \log(1/\delta)$ and $n = \tilde{O}(\text{poly}(\gamma^{-1})) \cdot \varepsilon^{-4}$. Then for any $m \geq m^*$, with probability at least $1 - \delta$ over the initialization and training sample, there is an iterate $k \in \{0, \ldots, K-1\}$ with $K = \tilde{O}(\text{poly}(\gamma^{-1})) \cdot \varepsilon^{-2}$ such that gradient descent with Gaussian initialization and step size $\eta = O(\gamma^4 \cdot m^{-1})$ satisfies

$$\mathbb{P}_{(x,y)\sim D}[y \cdot f_{W^{(k)}}(x) < 0] \leq \varepsilon.$$

This corollary shows that for deep residual networks, provided we have sufficient overparameterization, gradient descent is guaranteed to find networks that have arbitrarily high classification accuracy. In comparison with the results of Cao and Gu [5], the width $m$, number of samples $n$, step size $\eta$, and number of iterates $K$ required for the guarantees for residual networks given in Theorem 3.5 and Corollary 3.7 all have (at most) logarithmic dependence on $L$ as opposed to the exponential dependence in the corresponding results for the non-residual architecture. Additionally, we note that the step size and number of iterations required for our guarantees are independent of the depth, and this is due to the advantage of the residual architecture. Our analysis shows that the presence of skip connections in the network architecture removes the complications relating to the depth that traditionally arise in the analysis of non-residual architectures for a variety of reasons. The first is a technical one from the proof, in which we show that the Lipschitz constant of the network output

and the semismoothness of the network depend at most logarithmically on the depth, so that the network width does not blow up as the depth increases (see Lemmas 4.1 and 4.2 below). Second, the presence of skip-connections allows for representations that are learned in the first layer to be directly passed to later layers without needing to use a wider network to relearn those representations. This property was key to our proof of the gradient lower bound of Lemma 4.3 and has been used in previous approximation results for deep residual networks, e.g., Yarotsky [24].

## 4 Proof Sketch of the Main Theory

In this section we will provide a proof sketch of Theorems 3.5 and 3.6 and Corollary 3.7, following the proof technique of Cao and Gu [5]. We will first collect the key lemmas needed for their proofs, leaving the proofs of these lemmas for Appendix B. We shall assume throughout this section that the residual scaling parameter satisfies $\theta = 1/\Omega(L)$, which we note is a common assumption in the literature of residual network analysis [8, 1, 26].

Our first key lemma shows that the interlayer activations defined in (1) are uniformly bounded in $x$ and $l$ provided the network is sufficiently wide.

**Lemma 4.1** (Hidden layer and interlayer activations are bounded). Suppose that $W_1, \ldots, W_{L+1}$ are generated via Gaussian initialization. Then there exist absolute constants $C_0, C_1, C_2 > 0$ such that if $m \geq C_0 d \log(mL/\delta)$, then with probability at least $1 - \delta$, for any $l, l' = 1, \ldots, L+1$ with $l \leq l'$ and $x \in S^{d-1}$, we have $C_1 \leq \|x_l\|_2 \leq C_2$ and $\left\| H_l^{l'} \right\|_2 \leq C_2$.

Due to the scaling of $\theta$, we are able to get bounds on the interlayer and hidden layer activations that do not grow with $L$. As we shall see, this will be key for the sublinear dependence on $L$ for the results of Theorems 3.5 and 3.6. The fully connected architecture studied by Cao and Gu [5] had additional polynomial terms in $L$ for both upper bounds for $\|x_l\|_2$ and $\left\| H_l^{l'} \right\|_2$.

Our next lemma describes a semi-smoothness property of the neural network output $f_W$ and the empirical loss $L_S$.

**Lemma 4.2** (Semismoothness of network output and objective loss). Let $W_1, \ldots, W_{L+1}$ be generated via Gaussian initialization, and let $\tau \leq 1$. Define

$$h(\widehat{W}, \tilde{W}) := \left\| \widehat{W}_1 - \tilde{W}_1 \right\|_2 + \theta \sum_{l=2}^{L} \left\| \widehat{W}_l - \tilde{W}_l \right\|_2 + \left\| \widehat{W}_{L+1} - \tilde{W}_{L+1} \right\|_2.$$

There exist absolute constants $C, \overline{C} > 0$ such that if

$$m \geq C \left( \tau^{-\frac{2}{3}} (\log m)^{-1} \log(L/\delta) \vee \tau^{-\frac{4}{3}} d \log(m/(\tau\delta)) \vee d \log(mL/\delta) \right),$$

then with probability at least $1 - \delta$, we have for all $x \in S^{d-1}$ and $\widehat{W}, \tilde{W} \in \mathcal{W}(W, \tau)$,

$$f_{\widehat{W}}(x) - f_{\tilde{W}}(x) \leq \overline{C}\tau^{\frac{1}{3}} \sqrt{m \log m} \cdot h(\widehat{W}, \tilde{W}) + \overline{C}\sqrt{m} \cdot h(\widehat{W}, \tilde{W})^2$$
$$+ \sum_{l=1}^{L+1} \text{tr} \left[ \left( \widehat{W}_l - \tilde{W}_l \right)^\top \nabla_{W_l} f_{\tilde{W}}(x) \right].$$

and

$$L_S(\widehat{W}) - L_S(\tilde{W}) \leq \overline{C}\tau^{\frac{1}{3}} \sqrt{m \log m} \cdot h(\widehat{W}, \tilde{W}) \cdot \mathcal{E}_S(\tilde{W}) + \overline{C}m \cdot h(\widehat{W}, \tilde{W})^2$$
$$+ \sum_{l=1}^{L+1} \text{tr} \left[ \left( \widehat{W}_l - \tilde{W}_l \right)^\top \nabla_{W_l} L_S(\tilde{W}) \right].$$

The semismoothness of the neural network output function $f_W$ will be used in the analysis of generalization by Rademacher complexity arguments. For the objective loss $L_S$, we apply this lemma for weights along the trajectory of gradient descent. Since the difference in the weights of two consecutive steps of gradient descent satisfy $W_l^{(k+1)} - W_l^{(k)} = -\eta \nabla_{W_l} L_S(W^{(k)})$, the last term

in the bound for the objective loss $L_S$ will take the form $-\eta \sum_{l=1}^{L+1} \left\| \nabla_{W_l} L_S(W^{(k)}) \right\|_F^2$. Thus by simultaneously demonstrating (i) a lower bound for the gradient for at least one of the layers and (ii) an upper bound for the gradient at all layers (and hence an upper bound for $h(W^{(k+1)}, W^{(k)})$), we can connect the empirical surrogate loss $\mathcal{E}_S(W^{(k)})$ at iteration $k$ with that of the objective loss $L_S(W^{(k)})$ that will lead us to Theorem 3.5. Compared with the fully connected architecture of Cao and Gu [5], our bounds do not have any polynomial terms in $L$.

Thus the only remaining key items needed for our proof are upper bounds and lower bounds for the gradient of the objective loss, described in the following two lemmas.

**Lemma 4.3.** Let $W = (W_1, \ldots, W_{L+1})$ be weights at Gaussian initialization. There exist absolute constants $C, \underline{C}, \nu$ such that for any $\delta > 0$, provided $\tau \leq \nu \gamma^3$ and $m \geq C\gamma^{-2} \left( d \log \gamma^{-1} + \log(L/\delta) \right) \vee C \log(n/\delta)$, then with probability at least $1 - \delta$, for all $\tilde{W} \in \mathcal{W}(W, \tau)$, we have

$$\left\| \nabla_{W_{L+1}} L_S(\tilde{W}) \right\|_F^2 \geq \underline{C} \cdot m_{L+1} \cdot \gamma^4 \cdot \mathcal{E}_S(\tilde{W})^2.$$

**Lemma 4.4.** Let $W = (W_1, \ldots, W_{L+1})$ be weights at Gaussian initialization. There exists an absolute constant $C > 0$ such that for any $\delta > 0$, provided $m \geq C \left( d \vee \log(L/\delta) \right)$ and $\tau \leq 1$, we have for all $\tilde{W} \in \mathcal{W}(W, \tau)$ and all $l$,

$$\left\| \nabla_{W_l} L_S(\tilde{W}) \right\|_F \leq \theta^{\mathbb{1}(2 \leq l \leq L)} \cdot C\sqrt{m} \cdot \mathcal{E}_S(\tilde{W}).$$

Note that we provide only a lower bound for the gradient at the last layer. It may be possible to improve the degrees of the polynomial terms of the results in Theorems 3.5 and 3.6 by deriving lower bounds for the other layers as well.

With all of the key lemmas in place, we can proceed with a proof sketch of Theorems 3.5 and 3.6. The complete proofs can be found in Appendix A.

*Proof of Theorem 3.5.* Consider $h_k = h(W^{(k+1)}, W^{(k)})$, a quantity that measures the distance of the weights between gradient descent iterations. It takes the form

$$h_k = \eta \left[ \left\| \nabla_{W_1} L_S(W^{(k)}) \right\|_2 + \theta \sum_{l=2}^{L} \left\| \nabla_{W_l} L_S(W^{(k)}) \right\|_2 + \left\| \nabla_{W_{L+1}} L_S(W^{(k)}) \right\|_2 \right].$$

By Lemma 4.4 we can show that $h_k \leq C\eta \sqrt{m} \mathcal{E}_S(W^{(k)})$. The gradient lower bound in Lemma 4.3 substituted into Lemma 4.2 shows that the dominating term in the semismoothness comes from the gradient lower bound, so that we have for any $k$,

$$L_S(W^{(k+1)}) - L_S(W^{(k)}) \leq -C \cdot \eta \cdot m_{L+1} \cdot \gamma^4 \cdot \mathcal{E}_S(W^{(k)})^2.$$

We can telescope the above over $k$ to get a bound on the loss at iteration $k$ in terms of the bound on the r.h.s. and the loss at initialization. A simple concentration argument shows that the loss at initialization is small with mild overparameterization. By letting $k^* = \operatorname{argmin}_{[K-1]} \mathcal{E}_S(W^{(k)})^2$, we can thus show

$$\mathcal{E}_S(W^{(k^*)}) \leq C_3 \left( K\eta \cdot m \right)^{-\frac{1}{2}} \left( L_S(W^{(0)}) \right)^{\frac{1}{2}} \cdot \gamma^{-2} \leq C_3 \left( K\eta \cdot m \right)^{-\frac{1}{2}} \left( \log \frac{n}{\delta} \right)^{\frac{1}{4}} \cdot \gamma^{-2}.$$

$\square$

We provide below a proof sketch of the bound for the Rademacher complexity given in Theorem 3.6, leaving the rest for Appendix A.2.

*Proof of Theorem 3.6.* Let $\xi_i$ be independent Rademacher random variables. We consider a first-order approximation to the network output at initialization,

$$F_{W^{(0)}, W}(x) := f_{W^{(0)}}(x) + \sum_{l=1}^{L+1} \operatorname{tr} \left[ \left( W_l - W_l^{(0)} \right)^\top \nabla_{W_l} f_{W^{(0)}}(x) \right],$$

and bound the Rademacher complexity by two terms,

$$\widehat{\mathfrak{R}}_S[\mathcal{F}(W^{(0)}, \tau)] \le \mathbb{E}_\xi \left[ \sup_{W \in \mathcal{W}(W^{(0)}, \tau)} \frac{1}{n} \sum_{i=1}^n \xi_i [f(x_i) - F_{W^{(0)}, W}(x_i)] \right]$$

$$+ \mathbb{E}_\xi \left[ \sup_{W \in \mathcal{W}(W^{(0)}, \tau)} \frac{1}{n} \sum_{i=1}^n \xi_i \sum_{l=1}^{L+1} \mathrm{tr} \left[ \left( W_l - W_l^{(0)} \right)^\top \nabla_{W_l} f_{W^{(0)}}(x) \right] \right]$$

For the first term, taking $\tilde{W} = W^{(0)}$ in Lemma 4.2 results in $|f_W(x) - F_{W^{(0)}, W}(x)| \le C_3 \tau^{\frac{4}{3}} \sqrt{m \log m}$. For the second term, since $\|AB\|_F \le \|A\|_F \|B\|_2$, we reduce this term to a product of two terms. The first involves the norm of the distance of the weights from initialization, which is $\tau$. The second is the norm of the gradient at initialization, which can be taken care of by using Cauchy–Schwarz and the gradient formula (2) to get $\|\nabla_{W_l} f_{W^{(0)}}\|_F \le C_2 \theta^{\mathbb{1}(2 \le \ell \le L)} \sqrt{m}$. A standard application of Jensen inequality gives the $1/\sqrt{n}$ term. $\qquad \square$

Finally, we can put together Theorems 3.5 and 3.6 by appropriately choosing the scale of $\tau, \eta$, and $K$ to get Corollary 3.7. We leave the detailed algebraic calculations for Appendix A.3.

*Proof of Corollary 3.7.* We need only specify conditions on $\tau, \eta, K\eta$, and $m$ such that the results of Theorems 3.5 and 3.6 will hold, and making sure that each of the four terms in (3) are of the same scale. This can be satisfied by imposing the condition $K\eta = \nu'' \gamma^4 \tau^2 \left( \log(n/\delta) \right)^{-\frac{1}{2}}$ and

$$C_3 \left( K\eta m \right)^{-\frac{1}{2}} \left( \log(n/\delta) \right)^{\frac{1}{4}} \cdot \gamma^{-2} = C_2 \tau^{\frac{4}{3}} \sqrt{m \log m} = C_2 \tau \sqrt{m/n} = C_3 \sqrt{\log(1/\delta)/n} = \varepsilon/4.$$

$\qquad \square$

# 5 Conclusions

In this paper, we derived algorithm-dependent optimization and generalization results for overparameterized deep residual networks trained with random initialization using gradient descent. We showed that this class of networks is both small enough to ensure a small generalization gap and also large enough to achieve a small training loss. Important to our analysis is the insight that the introduction of skip connections allows for us to essentially ignore the depth as a complicating factor in the analysis, in contrast with the well-known difficulty of achieving nonvacuous generalization bounds for deep non-residual networks. This provides a theoretical understanding for the increased stability and generalization of deep residual networks over non-residual ones observed in practice.

## Acknowledgement

We would like to thank the anonymous reviewers for their helpful comments. This research was sponsored in part by the National Science Foundation IIS-1903202 and IIS-1906169. QG is also partially supported by the Salesforce Deep Learning Research Grant. The views and conclusions contained in this paper are those of the authors and should not be interpreted as representing any funding agencies.

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
