[Supplementary Material · resnet_supplementary_cameraready.pdf]

# A Proofs of Main Theorems and Corollaries

## A.1 Proof of Theorem 3.5

We first show that $W^{(k)} \in \mathcal{W}(W^{(0)}, \tau/2)$ for all $k \leq K$ satisfying $K\eta \leq \nu''\tau^2\gamma^4(\log(n/\delta))^{-1/2}$. Suppose $W^{(k')} \in \mathcal{W}(W^{(0)}, \tau/2)$ for all $k' = 1, \ldots, k-1$. By Lemma 4.4, we have

$$\left\|\nabla_{W_l} L_S(W^{(k')})\right\|_F \leq C_1 \theta^{\mathbb{1}(2 \leq l \leq L)} \sqrt{m} \cdot \mathcal{E}_S(W^{(k')}).$$

Since $\eta\sqrt{m} \leq \nu'\tau$ and $\mathcal{E}_S(\cdot) \leq 1$, we can make $\nu'$ small enough so that we have by the triangle inequality

$$\left\|W_l^{(k)} - W_l^{(0)}\right\|_F \leq \eta\left\|\nabla_{W_l} L_S(W^{(k-1)})\right\|_F + \frac{\tau}{2} \leq \tau. \tag{4}$$

Therefore we are in the $\tau$-neighborhood that allows us to apply the bounds described in the main section. Define

$$h_k := \eta\left[\left\|\nabla_{W_1} L_S(W^{(k)})\right\|_2 + \theta\sum_{l=2}^{L}\left\|\nabla_{W_l} L_S(W^{(k)})\right\|_2 + \left\|\nabla_{W_{L+1}} L_S(W^{(k)})\right\|_2\right].$$

Then using the upper bounds for the gradient given in Lemma 4.4, we have

$$h_k \leq \eta\left[C\sqrt{m}\mathcal{E}_S(W^{(k)}) + \theta\sum_{l=2}^{L}\left(\theta\sqrt{m}\mathcal{E}_S(W^{(k)})\right) + C\sqrt{m}\mathcal{E}_S(W^{(k)})\right] \leq C'\eta\sqrt{m}\mathcal{E}_S(W^{(k)}). \tag{5}$$

Notice that $h_k = h(W^{(k+1)}, W^{(k)})$ where $h$ is from Lemma 4.2. Hence, we have

$$L_S(W^{(k+1)}) - L_S(W^{(k)})$$

$$\leq C\tau^{\frac{1}{3}}\sqrt{m\log m} \cdot h_k \cdot \mathcal{E}_S(W^{(k)}) + Cmh_k^2 - \eta\sum_{l=1}^{L+1}\left\|\nabla_{W_l} L_S(W^{(k)})\right\|_F^2$$

$$\leq C\eta\tau^{\frac{1}{3}}\sqrt{m\log m} \cdot \sqrt{m} \cdot \mathcal{E}_S(W^{(k)})^2 + Cm^2\eta^2 \cdot \mathcal{E}_S(W^{(k)})^2 - C\eta \cdot m_{L+1} \cdot \gamma^4 \cdot \mathcal{E}_S(W^{(k)})^2$$

$$\leq \mathcal{E}_S(W^{(k)})^2 \cdot \left(C_1\eta\tau^{\frac{1}{3}}m\sqrt{\log m} + C_2 m^2 \cdot \eta^2 - C_3\eta \cdot m_{L+1} \cdot \gamma^4\right)$$

The first inequality follows by Lemma 4.2 and since $\operatorname{tr}(A^\top A) = \|A\|_F^2$. The second inequality uses the lower bound for the gradient given in Lemma 4.3 and (5). Therefore, if we take $\tau^{\frac{1}{3}}\sqrt{\log m} \leq \nu^{\frac{1}{3}}\gamma^4$, i.e. $\tau \leq \nu \cdot \gamma^{12}(\log m)^{-\frac{3}{2}}$ for some small enough constant $\nu$, and if we take $\eta \leq \nu' \cdot \gamma^4 m^{-1}$, then there is a constant $C > 0$ such that

$$L_S(W^{(k+1)}) - L_S(W^{(k)}) \leq -C \cdot \eta \cdot m_{L+1} \cdot \gamma^4 \cdot \mathcal{E}_S(W^{(k)})^2. \tag{6}$$

Re-writing this we have

$$\mathcal{E}_S(W^{(k)})^2 \leq C\gamma^{-4}(\eta m_{L+1})^{-1}\left(L_S(W^{(k)}) - L_S(W^{(k+1)})\right). \tag{7}$$

Before completing this part of the proof, we will need the following bound on the loss at initialization:

$$L_S(W^{(0)}) \leq C\sqrt{\log\frac{n}{\delta}}. \tag{8}$$

To see this, we notice that $f_W(x_i)$ is a sum of $m_{L+1}/2$ independent random variables (conditional on $x_{L,i}$),

$$f_W(x_i) = \sum_{j=1}^{m_{L+1}/2}\left[\sigma(w_{L+1,j}^\top x_{L,i}) - \sigma(w_{L+1,j+m_{L+1}/2}^\top x_{L,i})\right].$$

Applying the upper bound for $\|x_{L+1}\|_2$ given by Lemma 4.1 and Hoeffding inequality gives a constant $C_1 > 0$ such that with probability at least $1 - \delta$, $|f_{W^{(0)}}(x_i)| \leq C_1\sqrt{\log(n/\delta)}$ for all $i \in [n]$. Since $\ell(z) = \log(1 + \exp(-z)) \leq |z| + 1$ for all $z \in \mathbb{R}$, we have

$$L_S(W^{(0)}) = \frac{1}{n}\sum_{i=1}^{n}\ell(y_i \cdot f_{W^{(0)}}(x_i)) \leq 1 + C_1\sqrt{\log\frac{n}{\delta}} \leq C\sqrt{\log(n/\delta)}.$$

We can thus bound the distance from initialization by

$$\left\| W_l^{(k)} - W_l^{(0)} \right\|_F \le \eta \sum_{k'=0}^{k-1} \left\| \nabla_{W_l} L_S(W^{(k')}) \right\|_F$$

$$\le C\eta\sqrt{m} \sum_{k'=0}^{k-1} \mathcal{E}_S(W^{(k')})$$

$$\le C\eta\sqrt{m}\sqrt{k} \sqrt{\gamma^{-4} (\eta m_{L+1})^{-1} \sum_{k'=0}^{k-1} \left( L_S(W^{(k)}) - L_S(W^{(k+1)}) \right)}$$

$$\le C\sqrt{k\eta} \cdot \gamma^{-2} \left( \log \frac{n}{\delta} \right)^{\frac{1}{4}}$$

$$\le \frac{\tau}{2}.$$

The first line comes from the definition of gradient descent and the triangle inequality. For the second line, (4) allows us to apply Lemma 4.4. The third line follows by Cauchy–Schwarz and (7). The next line follows by (8), and the last since $k\eta \le \nu'' \tau^2 \gamma^4 (\log(n/\delta))^{-\frac{1}{2}}$. This completes the induction and shows that $W^{(k)} \in \mathcal{W}(W^{(0)}, \tau)$ for all $k \le K$.

For the second part of the proof, we want to derive an upper bound on the lowest empirical surrogate error over the trajectory of gradient descent. Since we have shown that $W^{(k)} \in \mathcal{W}(W^{(0)}, \tau/2)$ for $k \le K$, (6) and (8) both hold. Let $k^* = \operatorname{argmin}_{k \in \{0,\dots,K-1\}} \mathcal{E}_S(W^{(k)})^2$. Then telescoping (6) over $k$ yields

$$L_S(W^{(K)}) - L_S(W^{(0)}) \le -C \cdot \eta \cdot m_{L+1} \cdot \gamma^4 \cdot \sum_{k=1}^{K} \mathcal{E}_S(W^{(k)})^2$$

$$\le -C \cdot K\eta \cdot m_{L+1} \cdot \gamma^4 \cdot \mathcal{E}_S(W^{(k^*)})^2.$$

Rearranging the above gives

$$\mathcal{E}_S(W^{(k^*)}) \le C_3 (K\eta \cdot m)^{-\frac{1}{2}} \left( L_S(W^{(0)}) \right)^{\frac{1}{2}} \cdot \gamma^{-2} \le C_3 (K\eta \cdot m)^{-\frac{1}{2}} \left( \log \frac{n}{\delta} \right)^{\frac{1}{4}} \cdot \gamma^{-2},$$

where we have used that $L_S(\cdot)$ is always nonnegative in the first inequality and (8) in the second.

## A.2 Proof of Theorem 3.6

Denote $\mathcal{F}(W^{(0)}, \tau) = \{ f_W(x) : W \in \mathcal{W}(W^{(0)}, \tau) \}$, and recall the definition of the empirical Rademacher complexity,

$$\widehat{\mathfrak{R}}_S[\mathcal{F}(W^{(0)}, \tau)] = \mathbb{E}_\xi \left[ \sup_{f \in \mathcal{F}(W^{(0)}, \tau)} \frac{1}{n} \sum_{i=1}^{n} \xi_i f(x_i) \right] = \mathbb{E}_\xi \left[ \sup_{W \in \mathcal{W}(W^{(0)}, \tau)} \frac{1}{n} \sum_{i=1}^{n} \xi_i f(x_i) \right], \quad (9)$$

where $\xi = (\xi_1, \dots, \xi_n)^\top$ is an $n$-dimensional vector of i.i.d. $\xi_i \sim \operatorname{Unif}(\{\pm 1\})$. Since $y \in \{\pm 1\}$, $|\ell'(z)| \le 1$ and $\ell'(\cdot)$ is 1-Lipschitz, standard uniform convergence arguments (see, e.g., Shalev-Shwartz and Ben-David [21]) yield that with probability at least $1 - \delta$,

$$\sup_{W \in \mathcal{W}(W^{(0)}, \tau)} |\mathcal{E}_S(W) - \mathcal{E}_\mathcal{D}(W)| \le 2\mathbb{E}_S \widehat{\mathfrak{R}}_S \left[ \mathcal{F}(W^{(0)}, \tau) \right] + C_1 \sqrt{\frac{\log(1/\delta)}{n}}.$$

Since $-\ell'(x) = (1 + \exp(-x))^{-1}$ satisfies $-\ell'(x) < \frac{1}{2}$ if and only if $x < 0$, Markov's inequality gives

$$\mathbb{P}_{(x,y) \sim D} (y \cdot f_W(x) < 0) \le 2\mathbb{E}_{(x,y) \sim \mathcal{D}} (-\ell'(y \cdot f_W(x))) = 2\mathcal{E}_\mathcal{D}(W),$$

so that it suffices to get a bound for the empirical Rademacher complexity (9). If we define

$$F_{W^{(0)}, W}(x) := f_{W^{(0)}}(x) + \sum_{l=1}^{L+1} \operatorname{tr} \left[ \left( W_l - W_l^{(0)} \right)^\top \nabla_{W_l} f_{W^{(0)}}(x) \right],$$

then since $\sup_{a+b\in A+B}(a + b) \le \sup_{a\in A} a + \sup_{b\in B} b$, we have

$$\widehat{\mathfrak{R}}_S[\mathcal{F}(W^{(0)}, \tau)] \le \underbrace{\mathbb{E}_\xi \left[ \sup_{W\in\mathcal{W}(W^{(0)},\tau)} \frac{1}{n} \sum_{i=1}^n \xi_i[f(x_i) - F_{W^{(0)},W}(x_i)] \right]}_{I_1}$$

$$+ \underbrace{\mathbb{E}_\xi \left[ \sup_{W\in\mathcal{W}(W^{(0)},\tau)} \frac{1}{n} \sum_{i=1}^n \xi_i \sum_{l=1}^{L+1} \mathrm{tr} \left[ \left(W_l - W_l^{(0)}\right)^\top \nabla_{W_l} f_{W^{(0)}}(x) \right] \right]}_{I_2}$$

For the $I_1$ term, we take $\tilde{W} = W^{(0)}$ in Lemma 4.2 to get

$$|f_W(x) - F_{W^{(0)},W}(x)| \le C \left[ \tau^{\frac{4}{3}} \sqrt{m\log m}(2 + L\theta) \right] + C\tau^2 \sqrt{m}\,(2 + L\theta)$$

$$\le C\tau^{\frac{4}{3}} \sqrt{m\log m}.$$

For $I_2$, since $\|AB\|_F \le \|A\|_F \|B\|_2$, Lemma 4.1 yields for all $l$ and any matrix $\xi$,

$$\left\| x_l v^\top \cdot \xi \right\|_F \le \left\| x_l v^\top \right\|_F \|\xi\|_2 \le C\sqrt{m} \|\xi\|_2 .$$

Applying this to the gradient of $f$ at initialization given by (2) and using Lemma 4.1, there is a constant $C_2$ such that

$$\|\nabla_{W_l} f_{W^{(0)}}\|_F \le C_2 \theta^{\mathbb{1}(2\le l\le L)} \sqrt{m}. \tag{10}$$

We can therefore bound $I_2$ as follows:

$$I_2 \le \frac{\tau}{n} \sum_{l=1}^{L+1} \mathbb{E}_\xi \left\| \sum_{i=1}^n \xi_i \nabla_{W_l} f_{W^{(0)}}(x_i) \right\|_F$$

$$\le \frac{\tau}{n} \sum_{l=1}^{L+1} \sqrt{\mathbb{E} \left\| \sum_{i=1}^n \xi_i \nabla_{W_l} f_{W^{(0)}}(x_i) \right\|_F^2}$$

$$= \frac{\tau}{n} \sum_{l=1}^{L+1} \sqrt{\sum_{i=1}^n \|\nabla_{W_l} f_{W^{(0)}}(x_i)\|_F^2}$$

$$\le C\frac{\tau}{n} \left( \sqrt{nm} + \sum_{l=2}^L \sqrt{nm\theta^2} + \sqrt{nm} \right)$$

$$\le C\sqrt{\frac{m}{n}}\tau.$$

The first line above follows since $\mathrm{tr}(A^\top B) \le \|A\|_F \|B\|_F$ and $W \in \mathcal{W}(W^{(0)}, \tau)$. The second comes from Jensen inequality, with the third since $\xi_i^2 = 1$. The fourth line comes from (10), with the final inequality by the scale of $\theta$. This completes the proof.

### A.3 Proof of Corollary 3.7

We need only specify conditions on $\tau, \eta, K\eta$, and $m$ such that the results of Theorems 3.5 and 3.6 will hold, and such that each of the four terms in (3) are of the same scale $\varepsilon$. To get the two theorems to hold, we need $\tau \le \nu\gamma^{12} (\log m)^{-\frac{3}{2}}$, $\eta \le \nu'(\gamma^4 m^{-1} \wedge \tau m^{-\frac{1}{2}})$, $K\eta \le \nu''\tau^2\gamma^4 (\log(n/\delta))^{-\frac{1}{2}}$, and

$$m \ge C \left( \gamma^{-2} d\log\frac{1}{\gamma} \vee \gamma^{-2} \log\frac{L}{\delta} \vee d\log\frac{L}{\delta} \vee \tau^{-\frac{4}{3}} d\log\frac{L}{\tau\delta} \vee \tau^{-\frac{2}{3}} (\log m)^{-1} \log\frac{L}{\delta} \vee \log\frac{n}{\delta} \right).$$

We now find the appropriate scaling by first setting the upper bound for the surrogate loss given in Theorem 3.5 to $\varepsilon$ and then ensuring $\tau$ is such that the inequality required for $K\eta$ is satisfied:

$$C_3 (K\eta m)^{-\frac{1}{2}} (\log(n/\delta))^{\frac{1}{4}} \cdot \gamma^{-2} = \varepsilon, \qquad K\eta = \nu''\gamma^4\tau^2 (\log(n/\delta))^{-\frac{1}{2}} .$$

Substituting the values for $K\eta$ above, we get $C_4 m^{-\frac{1}{2}}\gamma^{-2}\tau^{-1}\sqrt{\log(n/\delta)} = \varepsilon$, so that

$$\tau = C_6\gamma^{-4}\varepsilon^{-1}m^{-\frac{1}{2}}\sqrt{\log(n/\delta)}. \tag{11}$$

Let $\widehat{m}$ be such that $\nu\gamma^{12}(\log m)^{-\frac{3}{2}} = \tau$, so that $m(\log m)^{-3} = C\nu^{-2}\gamma^{-32}(\log(n/\delta))\varepsilon^{-2}$. It is clear that such a $\widehat{m}$ can be written $\widehat{m} = \tilde{\Omega}(\text{poly}(\gamma^{-1})) \cdot \varepsilon^{-2}$. Finally we set

$$m^* = \max\left(\widehat{m}, d\log\frac{mL}{\delta}, \tau^{-\frac{4}{3}}\log\frac{m}{\tau\delta}\right).$$

By (11) we can write $\tau^{-\frac{4}{3}}\log(m/(\tau\delta)) = \gamma^{\frac{16}{3}}(\log(n/\delta))^{-\frac{2}{3}}\varepsilon^{\frac{4}{3}}m^{\frac{2}{3}}\log\left(m^{3/2}\gamma^4\varepsilon(\log(n/\delta))^{-\frac{1}{2}}/\delta\right)$. Thus we can take

$$m^* = \tilde{\Omega}(\text{poly}(\gamma^{-1})) \cdot \max(d, \varepsilon^{-2}) \cdot \log\frac{1}{\delta}.$$

Using (11) we see that $K = C\gamma^{-4}(\log(n/\delta))^{\frac{1}{2}}\varepsilon^{-2}$ and $\eta \leq \nu'\gamma^4 m^{-1}$ gives the desired forms of $K$ and $\eta$ as well as the first term of (3). For the second term of (3), we again use (11) to get $\tau^{\frac{4}{3}}\sqrt{m\log m} \leq C\gamma^{-\frac{16}{3}}(\log(n/\delta))^{\frac{2}{3}}\varepsilon^{-\frac{4}{3}}m^{-\frac{1}{6}} = R\varepsilon^{-\frac{4}{3}}m^{-\frac{1}{6}}$ where $R = \tilde{O}(\text{poly}(\gamma^{-1}))$. Since $\varepsilon^{-\frac{4}{3}}m^{-\frac{1}{6}} \leq \varepsilon$ iff $m \geq \varepsilon^{-14}$, this takes care of the second term in (3). For the third term, we again use (11) to write $\tau\sqrt{m/n} = C\gamma^{-4}\sqrt{\log(n/\delta)}n^{-\frac{1}{2}}\varepsilon^{-1} \leq \varepsilon$, which happens if $\sqrt{n/\log(n/\delta)} \geq C\varepsilon^{-2}\gamma^{-4}$, i.e., $n = \tilde{O}(\text{poly}(\gamma^{-1}))\varepsilon^{-4}$. For the final term of (3), it's clear that $\sqrt{\log(1/\delta)/n} \leq \varepsilon$ is satisfied when $n \geq C\varepsilon^{-2}\log(1/\delta)$, which is less stringent than the $\tilde{O}(\text{poly}(\gamma^{-1}))\varepsilon^{-4}$ requirement.

# B  Proofs of Key Lemmas

In this section we provide proofs to the key lemmas discussed in Section 4. We shall first provide the technical lemmas needed for their proof, and leave the proofs of the technical lemmas for Appendix C. Throughout this section, we assume that $\theta = 1/\Omega(L)$.

## B.1  Proof of Lemma 4.1: hidden and interlayer activations are bounded

We first recall a standard result from random matrix theory; see, e.g. Vershynin [23], Corollary 5.35.

**Lemma B.1.** Suppose $W_1, \ldots, W_{L+1}$ are generated by Gaussian initialization. Then there exist constants $C, C' > 0$ such that for any $\delta > 0$, if $m \geq d \vee C\log(L/\delta)$, then with probability at least $1 - \delta$, $\|W_l\|_2 \leq C'$ for all $l \in [L+1]$.

The next lemma bounds the spectral norm of the maps that the residual layers define. This is a key result that allows for the simplification of many of the arguments that are needed in non-residual architectures. Its proof is in Appendix C.1.

**Lemma B.2.** Suppose $W_1, \ldots, W_L$ are generated by Gaussian initialization. Then for any $\delta > 0$, there exist constants $C_0, C_0', C$ such that if $m \geq C_0\log(L/\delta)$, then with probability at least $1 - \delta$, for any $L \geq b \geq a \geq 2$, and for any tuple of diagonal matrices $\tilde{\Sigma}_a, \ldots, \tilde{\Sigma}_b$ satisfying $\left\|\tilde{\Sigma}_i\right\|_2 \leq 1$ for each $i = a, \ldots, b$, we have

$$\left\|(I + \theta\tilde{\Sigma}_b W_b^\top)(I + \theta\tilde{\Sigma}_{b-1}W_{b-1}^\top) \cdot \ldots \cdot (I + \theta\tilde{\Sigma}_a W_a^\top)\right\|_2 \leq \exp\left(C_0'\theta L\right) \leq 1.01. \tag{12}$$

In particular, if we consider $\tilde{\Sigma}_i = \Sigma_i(x)$ for any $x \in S^{d-1}$, we have with probability at least $1 - \delta$, for all $2 \leq a \leq b \leq L$ and for all $x \in S^{d-1}$,

$$\left\|(I + \theta\Sigma_b(x)W_b^\top)(I + \theta\Sigma_{b-1}(x)W_{b-1}^\top) \cdot \ldots \cdot (I + \theta\Sigma_a(x)W_a^\top)\right\|_2 \leq \exp\left(C_0'\theta L\right) \leq 1.01.$$

The next lemma we show concerns a Lipschitz property of the map $x \mapsto x_l$. Compared with the fully connected case, our Lipschitz constant does not involve any terms growing with $L$, which allows for the width dependence of our result to be only logarithmic in $L$. Its proof is in Appendix C.2.

**Lemma B.3.** Suppose $W_1, \ldots, W_L$ are generated by Gaussian initialization. There are constants $C, C' > 0$ such that for any $\delta > 0$, if $m \geq Cd\log(mL/\delta)$, then with probability at least $1 - \delta$, $\|x_l - x_l'\|_2 \leq C'\|x - x'\|_2$ for all $x, x' \in S^{d-1}$ and $l \in [L+1]$.

With the above technical lemmas in place, we can proceed with the proof of Lemma 4.1.

*Proof of Lemma 4.1.* We first show that a bound of the form $\underline{C} \leq \|\widehat{x}_l\|_2 \leq \overline{C}$ holds for all $\widehat{x}$ in an $\varepsilon$-net of $S^{d-1}$ and then use the Lipschitz property from Lemma B.3 to lift this result to all of $S^{d-1}$.

Let $\mathcal{N}^*$ be a $\tau_0$-net of $S^{d-1}$. By applying Lemma A.6 of Cao and Gu [5] to the first layer of our network, there exists a constant $C_1$ such that with probability at least $1 - \delta/3$, we can take $m = \Omega\left(d \log\left(m/(\tau_0 \delta)\right)\right)$ large enough so that

$$\|\widehat{x}_1\|_2 \leq 1 + C_1 \sqrt{\frac{d \log\left(m/(\tau_0 \delta)\right)}{m}} \leq 1.004.$$

If $2 \leq l \leq L$, by an application of Lemma B.2, by taking $m$ larger we have with probability at least $1 - \delta/3$, for all $2 \leq l \leq L, \widehat{x} \in \mathcal{N}^*$,

$$\begin{aligned}
\|\widehat{x}_l\|_2 &= \left\|(I + \theta \Sigma_l(\widehat{x}) W_l^\top) \cdots (I + \theta \Sigma_2(\widehat{x}) W_2^\top) \Sigma_1(\widehat{x}) W_1^\top \widehat{x}\right\|_2 \\
&\leq \left\|(I + \theta \Sigma_l(\widehat{x}) W_l^\top) \cdots (I + \theta \Sigma_2(\widehat{x}) W_2^\top)\right\|_2 \|\widehat{x}_1\|_2 \\
&\leq 1.01 \cdot \left(1 + C_1 \sqrt{\frac{d \log\left(m/(\tau_0 \delta)\right)}{m}}\right) \leq 1.015.
\end{aligned}$$

For the last fully connected layer, we can use a proof similar to that of Lemma A.6 in Cao and Gu [5] using the above upper bound on $\|\widehat{x}_L\|_2$ to get that with probability at least $1 - \delta$, for any $l \in [L+1]$ and $\widehat{x} \in \mathcal{N}^*$,

$$\|\widehat{x}_l\|_2 \leq 1.02. \tag{13}$$

For any $x \in S^{d-1}$, there exists $\widehat{x} \in \mathcal{N}^*$ such that $\|x - \widehat{x}\|_2 \leq \tau_0$. By Lemma B.3, this means that with probability at least $1 - \delta/2$, $\|x_l - \widehat{x}_l\|_2 \leq C_1 \tau_0$ for some $C_1 > 0$, and this holds over all $\widehat{x} \in \mathcal{N}^*$. Let $\tau_0 = 1/m$, so that $d \log\left(mL/(\tau_0 \delta)\right) \leq 2d \log(mL/\delta)$. Then (13) yields that with probability at least $1 - \delta$, for all $x \in S^{d-1}$ and all $l \in [L+1]$,

$$\|x_l\|_2 \leq \|\widehat{x}_l\|_2 + \|x_l - \widehat{x}_l\|_2 \leq 1.02 + C_1/m \leq 1.024.$$

As for the lower bound, we again let $\mathcal{N}^*$ be an arbitrary $\tau_0$-net of $S^{d-1}$. For $l = 1$, we use Lemma A.6 in Cao and Gu [5] to get constants $C, C'$ such that provided $m \geq Cd \log\left(m/(\tau_0 \delta)\right)$, then we have with probability at least $1 - \delta/3$, for all $\widehat{x} \in \mathcal{N}^*$,

$$\|\widehat{x}_l\|_2 \geq 1 - C' \sqrt{dm^{-1} \log\left(3m/(\tau_0 \delta)\right)} \qquad (l = 1, 2, \ldots, L). \tag{14}$$

To see that the above holds for layers $2 \leq l \leq L$, we note that it deterministically holds that $\widehat{x}_{l,j} \geq \widehat{x}_{1,j}$ for such $l$ and all $j$. For the final layer, we follow a proof similar to Lemma A.6 of Cao and Gu [5] with an application of (13) to get that with probability at least $1 - \delta/3$,

$$\|\widehat{x}_{L+1}\|_2^2 \geq \|\widehat{x}_L\|_2^2 - C_3 \sqrt{dm^{-1} \log\left(3/(\tau_0 \delta)\right)}.$$

Thus $m = \Omega(d \log(m/(\tau_0 \delta)))$ implies there is a constant $C_4$ such that with probability at least $1 - \delta$, for all $l \in [L+1]$ and $\widehat{x} \in \mathcal{N}^*$,

$$\|\widehat{x}_l\|_2 \geq C_4 > 0. \tag{15}$$

By Lemma B.3, we have with probability at least $1 - \delta$, for all $x \in S^{d-1}$,

$$\|x_l\|_2 \geq \|\widehat{x}_l\|_2 - \|x_l - \widehat{x}_l\|_2 \geq C_4 - C_1 \tau_0.$$

Thus by taking $\tau_0$ to be a sufficiently small universal constant, we get the desired lower bound.

We now demonstrate the upper bound for $\left\|H_l^{l'}\right\|_2$. Since $H_l^{l'} = x_{l'}$ when $l = 1$, we need only consider the case $l > 1$. If $l' \leq L$, then $H_l^{l'}$ appears in the bound for Lemma B.2 and so we are done. For $l' = L + 1$, by Lemmas B.1 and B.2 we have

$$\begin{aligned}
\left\|H_l^{L+1}\right\|_2 &= \left\|\Sigma_{L+1}(x) W_{L+1}^\top \prod_{r=l}^{L} \left(I + \theta \Sigma_r(x) W_r^\top\right)\right\|_2 \\
&\leq \|\Sigma_{L+1}(x)\|_2 \|W_{L+1}\|_2 \left\|\prod_{r=l}^{L} \left(I + \theta \Sigma_r(x) W_r^\top\right)\right\|_2 \leq C.
\end{aligned}$$

$\square$

## B.2 Proof of Lemma 4.2: semismoothness

To prove the semismoothness result, we need two technical lemmas. The first lemma concerns a Lipschitz-type property with respect to the weights, along with a characterization of the changing sparsity patterns of the rectifier activations at each layer. The second lemma characterizes how the neural network output behaves if we know that one of the initial layers has a given sparsity pattern. This allows us to develop the desired semi-smoothness even though ReLU is non-differentiable. The proof for Lemmas B.4 and B.5 can be found in Appendix C.3 and C.4, respectively.

**Lemma B.4.** Let $W = (W_1, \ldots, W_{L+1})$ be generated by Gaussian initialization, and let $\widehat{W} = (\widehat{W}_1, \ldots, \widehat{W}_{L+1}), \tilde{W} = (\tilde{W}_1, \ldots, \tilde{W}_{L+1})$ be weight matrices such that $\widehat{W}, \tilde{W} \in \mathcal{W}(W, \tau)$. For $x \in S^{d-1}$, let $\Sigma_l(x), \widehat{\Sigma}_l(x), \tilde{\Sigma}_l(x)$ and $x_l, \widehat{x}_l, \tilde{x}_l$ be the binary matrices and hidden layer outputs of the $l$-th layers with parameters $W, \widehat{W}, \tilde{W}$ respectively. There exist absolute constants $C_1, C_2, C_3$ such that for any $\delta > 0$, if $m \geq C_1 \tau^{-\frac{4}{3}} \cdot d \log(m/(\tau\delta)) \vee C_1 d \log(mL/\delta)$, then with probability at least $1 - \delta$, for any $x \in S^{d-1}$ and any $l \in [L+1]$, we have

$$
\|\widehat{x}_l - \tilde{x}_l\|_2 \leq \begin{cases} C_2 \left\|\widehat{W}_1 - \tilde{W}_1\right\|_2, & l = 1, \\ C_2 \left\|\widehat{W}_1 - \tilde{W}_1\right\|_2 + \theta C_2 \sum_{r=2}^{l} \left\|\widehat{W}_r - \tilde{W}_r\right\|_2, & 2 \leq l \leq L, \\ C_2 \left\|\widehat{W}_1 - \tilde{W}_1\right\|_2 + \theta C_2 \sum_{r=2}^{L} \left\|\widehat{W}_r - \tilde{W}_r\right\|_2 + C_2 \left\|\widehat{W}_{L+1} - \tilde{W}_{L+1}\right\|_2, & l = L+1. \end{cases}
$$

and

$$
\left\|\widehat{\Sigma}_l(x) - \tilde{\Sigma}_l(x)\right\|_0 \leq C_3 m \tau^{\frac{2}{3}}.
$$

**Lemma B.5.** Let $W_1, \ldots, W_{L+1}$ be generated by Gaussian initialization. Let $\tilde{W}_l$ be such that $\left\|W_l - \tilde{W}_l\right\|_2 \leq \tau$ for all $l$, and let $\tilde{\Sigma}_l(x)$ be the diagonal activation matrices corresponding to $\tilde{W}_l$, and $\tilde{H}_l^{l'}(x)$ the corresponding interlayer activations defined in (1). Suppose that $\left\|\tilde{\Sigma}_l(x) - \Sigma_l(x)\right\|_0 \leq s$ for all $x \in S^{d-1}$ and all $l$. Define, for $l \geq 2$ and $a \in \mathbb{R}^{m_{l-1}}$,

$$
g_l(a, x) := v^\top \tilde{H}_l^{L+1}(x) a.
$$

Then there exists a constant $C > 0$ such that for any $\delta > 0$, provided $m \geq C\tau^{-\frac{2}{3}}(\log m)^{-1} \log(L/\delta)$, we have with probability at least $1 - \delta$ and all $2 \leq l \leq L+1$,

$$
\sup_{\|x\|_2 = \|a\|_2 = 1, \|a\|_0 \leq s} |g_l(a, x)| \leq C_1 \left[\tau\sqrt{m} + \sqrt{s \log m}\right].
$$

In comparison with the fully connected case of Cao and Gu [5], our bounds in Lemmas B.4 and B.5 do not involve polynomial terms in $L$, and the residual scaling $\theta$ further scales the dependence of the hidden layer activations on the intermediate layers.

With the above two technical lemmas, we can proceed with the proof of Lemma 4.2.

*Proof of semismoothness, Lemma 4.2.* Recalling the notation of interlayer activations $H_l^{l'}$ from (1), we have for any $l \in [L+1]$ $f_{\widehat{W}}(x) = v^\top \widehat{H}_{l+1}^{L+1} \widehat{x}_l$, where we have denoted $H_l^{l'}(x) = H_l^{l'}$ for notational simplicity. Similarly, in what follows we denote $\Sigma(x)$ by $\Sigma$ with the understanding that each diagonal matrix $\Sigma$ still depends on $x$. We have the decomposition

$$
\widehat{H}_2^{L+1}\widehat{\Sigma}_1\widehat{W}_1 x = \left(\widehat{H}_2^{L+1} - \tilde{H}_2^{L+1}\right)\widehat{\Sigma}_1\widehat{W}_1^\top x + \tilde{H}_2^{L+1}\widehat{\Sigma}_1\widehat{W}_1^\top x,
$$

and for $2 \leq l \leq L$,

$$
\widehat{H}_l^{L+1} - \tilde{H}_l^{L+1} = \left(\widehat{H}_{l+1}^{L+1} - \tilde{H}_{l+1}^{L+1}\right)\left(I + \theta\widehat{\Sigma}_l\widehat{W}_l^\top\right) + \theta\tilde{H}_{l+1}^{L+1}\left(\widehat{\Sigma}_l\widehat{W}_l^\top - \tilde{\Sigma}_l\tilde{W}_l^\top\right).
$$

Thus we can write

$$\widehat{H}_1^{L+1}(x) - \tilde{H}_1^{L+1}(x) = \left(\widehat{H}_2^{L+1} - \tilde{H}_2^{L+1}\right)\widehat{\Sigma}_1\widehat{W}_1^\top x + \tilde{H}_2^{L+1}\left(\widehat{\Sigma}_1\widehat{W}_1^\top - \tilde{\Sigma}_1\tilde{W}_1^\top\right)x$$

$$= \left(\widehat{\Sigma}_{L+1}\widehat{W}_{L+1}^\top - \tilde{\Sigma}_{L+1}\tilde{W}_{L+1}^\top\right)\widehat{x}_L$$

$$+ \theta\sum_{l=2}^L \tilde{H}_{l+1}^{L+1}\left(\widehat{\Sigma}_l\widehat{W}_l^\top - \tilde{\Sigma}_l\tilde{W}_l^\top\right)\widehat{x}_{l-1} + \tilde{H}_2^{L+1}\left(\widehat{\Sigma}_1\widehat{W}_1 - \tilde{\Sigma}_1\tilde{W}_1\right)x.$$

We thus want to bound the quantity

$$f_{\widehat{W}}(x) - f_{\tilde{W}}(x) = v^\top\left(\widehat{\Sigma}_{L+1}\widehat{W}_{L+1}^\top - \tilde{\Sigma}_{L+1}\tilde{W}_{L+1}^\top\right)\widehat{x}_L \qquad (T_1)$$

$$+ \theta v^\top\left[\sum_{l=2}^L \tilde{H}_{l+1}^{L+1}\left(\widehat{\Sigma}_l\widehat{W}_l^\top - \tilde{\Sigma}_l\tilde{W}_l^\top\right)\widehat{x}_{l-1}\right] \qquad (T_2)$$

$$+ v^\top\left[\tilde{H}_2^{L+1}\left(\widehat{\Sigma}_1\widehat{W}_1 - \tilde{\Sigma}_1\tilde{W}_1\right)x\right]. \qquad (T_3) \qquad (16)$$

We deal with the three terms separately. The idea in each is the same.

**First term, $T_1$.** We write this as the sum of three terms $v^\top(I_1 + I_2 + I_3)$, where

$$\left(\widehat{\Sigma}_{L+1}\widehat{W}_{L+1}^\top - \tilde{\Sigma}_{L+1}\tilde{W}_{L+1}^\top\right)\widehat{x}_L$$

$$= \underbrace{\left(\widehat{\Sigma}_{L+1} - \tilde{\Sigma}_{L+1}\right)\widehat{W}_{L+1}^\top\widehat{x}_L}_{I_1} + \underbrace{\tilde{\Sigma}_{L+1}\left(\widehat{W}_{L+1}^\top - \tilde{W}_{L+1}^\top\right)(\widehat{x}_L - \tilde{x}_L)}_{I_2} + \underbrace{\tilde{\Sigma}_{L+1}\left(\widehat{W}_{L+1}^\top - \tilde{W}_{L+1}^\top\right)\tilde{x}_L}_{I_3}.$$

$$(17)$$

By directly checking the signs of the diagonal matrices, we can see that for any $l = 1, \ldots, L+1$,

$$\left\|\left(\widehat{\Sigma}_l - \tilde{\Sigma}_l\right)\widehat{W}_l^\top\widehat{x}_{l-1}\right\|_2 \le C_1\left\|\widehat{W}_l - \tilde{W}_l\right\|_2 + C_1\|\widehat{x}_{l-1} - \tilde{x}_{l-1}\|_2. \qquad (18)$$

We will use Lemma B.4 to get specific bounds for each $l$. Denote $|\Sigma|$ as the entrywise absolute values of a diagonal matrix $\Sigma$, so that $|\Sigma|\Sigma = \Sigma$ provided the diagonal entries are all in $\{0, \pm 1\}$. Then we can write

$$|v^\top I_1| = \left\|v^\top\left|\widehat{\Sigma}_{L+1} - \tilde{\Sigma}_{L+1}\right|\left(\widehat{\Sigma}_{L+1} - \tilde{\Sigma}_{L+1}\right)\widehat{W}_{L+1}^\top\widehat{x}_L\right\|_2$$

$$\le C_3\tau^{\frac{1}{3}}\sqrt{m}\left\|\left(\widehat{\Sigma}_{L+1} - \tilde{\Sigma}_{L+1}\right)\widehat{W}_{L+1}^\top\widehat{x}_L\right\|_2$$

$$\le C_3\tau^{\frac{1}{3}}\sqrt{m}\cdot\left(C_1\left\|\widehat{W}_{L+1} - \tilde{W}_{L+1}\right\|_2 + C_1\|\widehat{x}_L - \tilde{x}_L\|_2\right) \qquad (19)$$

The first inequality follows by first noting that for any vector $a$ with $|a_i| \le 1$ it holds that $\left\|v^\top a\right\|_2 \le \|a\|_0^{\frac{1}{2}}$, and then applying Lemma B.4 to get $\left\|\widehat{\Sigma}_{L+1} - \tilde{\Sigma}_{L+1}\right\|_0 \le s = O\left(m\tau^{\frac{2}{3}}\right)$. The last line is by (18).

The $I_2$ term in (17) follows from a simple application of Cauchy–Schwarz:

$$|v^\top I_2| \le \sqrt{m}\cdot C\cdot\left\|\widehat{W}_{L+1} - \tilde{W}_{L+1}\right\|_2\|\widehat{x}_L - \tilde{x}_L\|_2. \qquad (20)$$

Putting together (19) and (20) shows that we can bound $T_1$ in (16) by

$$T_1 \le C_3\tau^{\frac{1}{3}}\sqrt{m}\cdot\left(C_1\left\|\widehat{W}_{L+1} - \tilde{W}_{L+1}\right\|_2 + C_1\|\widehat{x}_L - \tilde{x}_L\|_2\right) + \sqrt{m}\cdot C\cdot\left\|\widehat{W}_{L+1} - \tilde{W}_{L+1}\right\|_2\|\widehat{x}_L - \tilde{x}_L\|_2$$

$$+ v^\top\tilde{\Sigma}_{L+1}\left(\widehat{W}_{L+1} - \tilde{W}_{L+1}\right)^\top\tilde{x}_L$$

$$\le C_3\tau^{\frac{1}{3}}\sqrt{m}\left(C_1\left\|\widehat{W}_{L+1} - \tilde{W}_{L+1}\right\|_2 + C_1'\left[\left\|\widehat{W}_1 - \tilde{W}_1\right\|_2 + \theta\sum_{r=2}^L\left\|\widehat{W}_r - \tilde{W}_r\right\|_2\right]\right)$$

$$+ C\sqrt{m}\left\|\widehat{W}_{L+1} - \tilde{W}_{L+1}\right\|_2\left(\left\|\widehat{W}_1 - \tilde{W}_1\right\|_2 + \theta\sum_{r=2}^L\left\|\tilde{W}_r - \widehat{W}_r\right\|_2\right)$$

$$+ v^\top\tilde{\Sigma}_{L+1}\left(\widehat{W}_{L+1} - \tilde{W}_{L+1}\right)^\top\tilde{x}_L. \qquad (21)$$

**Second term, $T_2$.** We again use a decomposition like (17):

$$\tilde{H}_{l+1}^{L+1} \left( \widehat{\Sigma}_l \widehat{W}_l^\top - \tilde{\Sigma}_l \tilde{W}_l^\top \right) \widehat{x}_{l-1}$$
$$= \underbrace{\tilde{H}_{l+1}^{L+1} \left( \widehat{\Sigma}_l - \tilde{\Sigma}_l \right) \widehat{W}_l^\top \widehat{x}_{l-1}}_{I_1} + \underbrace{\tilde{H}_{l+1}^{L+1} \tilde{\Sigma}_l \left( \widehat{W}_l^\top - \tilde{W}_l^\top \right) (\widehat{x}_{l-1} - \tilde{x}_{l-1})}_{I_2} + \underbrace{\tilde{H}_{l+1}^{L+1} \tilde{\Sigma}_l \left( \widehat{W}_l^\top - \tilde{W}_l^\top \right) \tilde{x}_{l-1}}_{I_3}.$$

$$(22)$$

For $I_1$, we note that Lemma B.4 gives sparsity level $s = O(m\tau^{\frac{2}{3}})$ for $\widehat{\Sigma}_l - \tilde{\Sigma}_l$. We thus proceed similarly as for the term $T_1$ to get

$$|v^\top I_1| \leq \left\| v^\top \tilde{\Sigma}_{L+1} \tilde{W}_{L+1}^\top \tilde{H}_{l+1}^L \left| \widehat{\Sigma}_l - \tilde{\Sigma}_l \right| \left( \widehat{\Sigma}_l - \tilde{\Sigma}_l \right) \widehat{W}_l^\top \widehat{x}_{l-1} \right\|_2$$
$$\leq C\tau^{\frac{1}{3}} \sqrt{m \log m} \cdot \left( C_1 \left\| \widehat{W}_l - \tilde{W}_l \right\|_2 + C_2 \left\| \widehat{x}_{l-1} - \tilde{x}_{l-1} \right\|_2 \right).$$

The above follows since $s \log m \geq C \log(L/\delta)$ holds for $s = m\tau^{\frac{2}{3}}$, and we can hence apply Lemma B.5 and (18). The bound for the $I_2$ term again follows by Cauchy–Schwarz,

$$|v^\top I_2| \leq \sqrt{m} \cdot C \cdot \left\| \widehat{W}_l - \tilde{W}_l \right\|_2 \left\| \widehat{x}_{l-1} - \tilde{x}_{l-1} \right\|_2.$$

Thus, for the term $T_2$ in (16) we have

$$T_2 \leq \theta \sum_{l=2}^{L} \left( C_6 \tau^{\frac{1}{3}} \sqrt{m \log m} \left\| \widehat{W}_l - \tilde{W}_l \right\|_2 + C\tau^{\frac{1}{3}} \sqrt{m \log m} \left\| \widehat{W}_1 - \tilde{W}_1 \right\|_2 \right)$$
$$+ \theta^2 \sum_{l=2}^{L} \left( \tau^{\frac{1}{3}} \sqrt{m \log m} \sum_{r=2}^{l} \left\| \tilde{W}_r - \widehat{W}_r \right\|_2 \right)$$
$$+ \theta \sum_{r=2}^{L} \sqrt{m} C \left\| \widehat{W}_l - \tilde{W}_l \right\|_2 \left( \left\| \widehat{W}_1 - \tilde{W}_1 \right\|_2 + \theta \sum_{r=l}^{2} \left\| \widehat{W}_r - \tilde{W}_r \right\|_2 \right)$$
$$+ \theta \sum_{l=2}^{L} v^\top \tilde{H}_{l+1}^{L+1} \tilde{\Sigma}_l \left( \widehat{W}_l^\top - \tilde{W}_l^\top \right) \tilde{x}_{l-1}. \tag{23}$$

**Third term, $T_3$.** For $T_3$, we work on the quantity

$$\tilde{H}_2^{L+1} \left( \widehat{\Sigma}_1 \widehat{W}_1^\top - \tilde{\Sigma}_1 \tilde{W}_1^\top \right) x = \tilde{H}_2^{L+1} \left( \widehat{\Sigma}_1 - \tilde{\Sigma}_1 \right) \widehat{W}_1^\top x + \tilde{H}_2^{L+1} \tilde{\Sigma}_1 \left( \widehat{W}_1 - \tilde{W}_1 \right) x.$$

Thus, we again have by Lemma B.5,

$$T_3 \leq \left\| v^\top \tilde{H}_2^{L+1} \left| \widehat{\Sigma}_1 - \tilde{\Sigma}_1 \right| \right\|_2 \left\| \left( \widehat{\Sigma}_1 - \tilde{\Sigma}_1 \right) \widehat{W}_1 x \right\|_2 + v^\top \tilde{H}_2^{L+1} \tilde{\Sigma}_1 \left( \widehat{W}_1 - \tilde{W}_1 \right) x$$
$$\leq \tau^{\frac{1}{3}} \sqrt{m \log m} \left\| \widehat{W}_1 - \tilde{W}_1 \right\|_2 + v^\top \tilde{H}_2^{L+1} \tilde{\Sigma}_1 \left( \widehat{W}_1 - \tilde{W}_1 \right) x. \tag{24}$$

Using the linearity of the trace operator and that $\text{tr}(ABC) = \text{tr}(CAB) = \text{tr}(BCA)$ for any matrices $A, B, C$ for which those products are defined, we can use the gradient formula (2) to calculate for any $l \in [L+1]$,

$$\theta^{\mathbb{1}(2 \leq l \leq L)} v^\top \tilde{H}_l^{L+1} \tilde{\Sigma}_l \left( \widehat{W}_l - \tilde{W}_l \right)^\top \tilde{x}_{l-1} = \text{tr} \left[ \left( \widehat{W}_l - \tilde{W}_l \right)^\top \nabla_{W_l} f_{\tilde{W}}(x) \right]. \tag{25}$$

Let now

$$h(\widehat{W}, \tilde{W}) := \left\| \widehat{W}_1 - \tilde{W}_1 \right\|_2 + \theta \sum_{l=2}^{L} \left\| \widehat{W}_l - \tilde{W}_l \right\|_2 + \left\| \widehat{W}_{L+1} - \tilde{W}_{L+1} \right\|_2.$$

Substituting the bounds from (21), (23), (24) and (25) thus yield for some constant $\overline{C}$,

$$f_{\widehat{W}}(x) - f_{\tilde{W}}(x) \leq C\tau^{\frac{1}{3}}\sqrt{m\log m}\left[\left\|\widehat{W}_1 - \tilde{W}_1\right\|_2 + \theta C\sum_{l=2}^{L}\left\|\widehat{W}_l - \tilde{W}_l\right\|_2 + C\left\|\widehat{W}_{L+1} - \tilde{W}_{L+1}\right\|_2\right]$$

$$+ C\tau^{\frac{1}{3}}\sqrt{m\log m}\left[\left\|\widehat{W}_1 - \tilde{W}_1\right\|_2 + C\left\|\widehat{W}_1 - \tilde{W}_1\right\|_2 + \theta C\sum_{l=2}^{l}\left\|\widehat{W}_l - \tilde{W}_l\right\|_2\right]$$

$$+ C\sqrt{m}\left[\left\|\widehat{W}_{L+1} - \tilde{W}_{L+1}\right\|_2 \cdot \left\|\widehat{W}_1 - \tilde{W}_{L+1}\right\|_2 + \theta\left\|\widehat{W}_{L+1} - \tilde{W}_{L+1}\right\|_2\sum_{r=2}^{L}\left\|\widehat{W}_r - \tilde{W}_r\right\|_2\right.$$

$$\left. + \theta\sum_{l=2}^{L}\left\|\widehat{W}_l - \tilde{W}_l\right\|_2\left\|\widehat{W}_1 - \tilde{W}_1\right\|_2 + \theta\sum_{l=2}^{L}\left\|\widehat{W}_l - \tilde{W}_l\right\|_2 \cdot \left(\theta\sum_{r=2}^{l}\left\|\widehat{W}_r - \tilde{W}_r\right\|_2\right)\right]$$

$$+ \sum_{l=1}^{L+1}\mathrm{tr}\left[\left(\widehat{W}_l - \tilde{W}_l\right)\nabla_{W_l}f_{\tilde{W}}(x)\right]$$

$$\leq \overline{C}\tau^{\frac{1}{3}}\sqrt{m\log m}\cdot h(\widehat{W}, \tilde{W}) + \overline{C}\sqrt{m}\cdot h(\widehat{W}, \tilde{W})^2 + \sum_{l=1}^{L+1}\mathrm{tr}\left[\left(\widehat{W}_l - \tilde{W}_l\right)\nabla_{W_l}f_{\tilde{W}}(x)\right] \quad (26)$$

This completes the proof of semi-smoothness of $f_W$. For $L_S$, denote $\widehat{y}_i, \tilde{y}_i$ as the outputs of the network for input $x_i$ under weights $\widehat{W}, \tilde{W}$ respectively. Since $\ell''(z) \leq 0.5$ for all $z \in \mathbb{R}$, if we denote $\Delta_i = \widehat{y}_i - \tilde{y}_i = f_{\widehat{W}}(x_i) - f_{\tilde{W}}(x_i)$, we have

$$L_S(\widehat{W}) - L_S(\tilde{W}) \leq \frac{1}{n}\sum_{i=1}^{n}\left[\ell'(y_i\tilde{y}_i)\cdot y_i \cdot \Delta_i + \frac{1}{4}\Delta_i^2\right].$$

Applying (26) and using that $-n^{-1}\sum_{i=1}^{n}\ell'(z_i) \leq 1$ for any $z_i \in \mathbb{R}$,

$$\frac{1}{n}\sum_{i=1}^{n}\ell'(y_i\tilde{y}_i)y_i\cdot\Delta_i \leq C\tau^{\frac{1}{3}}\sqrt{m\log m}\cdot h(\widehat{W}, \tilde{W})\cdot\mathcal{E}_S(\tilde{W}) + C\sqrt{m}\cdot h(\widehat{W}, \tilde{W})^2\cdot\mathcal{E}_S(\tilde{W})$$

$$+ \sum_{l=1}^{L+1}\frac{1}{n}\sum_{i=1}^{n}\ell'(y_i\tilde{y}_i)\cdot y_i\cdot\mathrm{tr}\left[\left(\widehat{W}_l - \tilde{W}_l\right)\nabla_{W_l}f_{\tilde{W}}(x_i)\right].$$

Linearity of the trace operator allows the last term in the above display to be written as

$$\sum_{l=1}^{L+1}\mathrm{tr}\left[\left(\widehat{W}_l - \tilde{W}_l\right)\nabla_{W_l}L_S(\tilde{W})\right].$$

Moreover, using Lemma B.4,

$$\Delta_i^2 = \left[v^\top(\widehat{x}_{L+1,i} - \tilde{x}_{L+1,i})\right]^2 \leq \|v\|_2^2\|\widehat{x}_{L+1,i} - \tilde{x}_{L+1,i}\|_2^2 \leq C_2\cdot m\cdot h(\widehat{W}, \tilde{W})^2.$$

This term dominates the corresponding $h^2$ term coming from $\Delta_i$ and so completes the proof.

$\square$

## B.3 Proof of Lemma 4.3: gradient lower bound

This is the part of the proof that makes use of the assumption on the data distribution given in Assumption 3.2, and is key to the mild overparameterization required for our generalization result. The key technical lemma needed for the proof of the gradient lower bound is given below. The proof of Lemma B.6 can be found in Appendix C.5.

**Lemma B.6.** Let $a(x, y) : S^{d-1} \times \{\pm 1\} \to [0, 1]$. For any $\delta > 0$, there is a constant $C > 0$ such that if $m \geq C\gamma^{-2}(d\log(1/\gamma) + \log(L/\delta))$ and $m \geq C\log(n/\delta)$ then for any such function $a$, we have with probability at least $1 - \delta$,

$$\sum_{j=1}^{m_{L+1}}\left\|\frac{1}{n}\sum_{i=1}^{n}\left[a(x_i, y_i)\cdot y_i\cdot\sigma'\left(w_{L+1,j}^\top x_{L,i}\right)\cdot x_{L,i}\right]\right\|_2^2 \geq \frac{1}{67}m_{L+1}\gamma^2\left(\frac{1}{n}\sum_{i=1}^{n}a(x_i, y_i)\right)^2.$$

*Proof of Lemma 4.3.* Let $\tilde{y}_i := f_{\tilde{W}}(x_i)$, and define $g_j :=$ $\frac{1}{n}\sum_{i=1}^{n}\left[\ell'(y_i\tilde{y}_i)\cdot v_j\cdot y_i\cdot\sigma'(w_{L+1,j}^\top x_{L,i})\cdot x_{L,i}\right]$ so that

$$\sum_{j=1}^{m_{L+1}}\|g_j\|_2^2 = \sum_{j=1}^{m_{L+1}}\left\|\frac{1}{n}\sum_{i=1}^{n}\left[\ell'(y_i\tilde{y}_i)\cdot y_i\cdot\sigma'(w_{L+1,j}^\top x_{L,i})\cdot x_{L,i}\right]\right\|_2^2.$$

Recall that $\mathcal{E}_S(\tilde{W}) = -n^{-1}\sum_{i=1}^{n}\ell'(y_i\tilde{y}_i)$. Applying Lemma B.6 gives

$$\sum_{j=1}^{m_{L+1}}\|g_j\|_2^2 \geq \frac{1}{67}m_{L+1}\gamma^2[\mathcal{E}_S(\tilde{W})]^2. \tag{27}$$

By Lemma 4.1, for any $j \in [m_{L+1}]$, we have

$$\|g_j\|_2 \leq \frac{1}{n}\sum_{i=1}^{n}\left\|\ell'(y_i\tilde{y}_i)\cdot v_j\cdot y_i\cdot\sigma'(w_{L+1,j}^\top x_{L,i})\cdot x_{L,i}\right\|_2 \leq 1.02\mathcal{E}_S(\tilde{W}). \tag{28}$$

Define

$$A := \left\{j \in [m_{L+1}] : \|g_j\|_2^2 \geq \frac{1}{2\cdot 67}\gamma^2\left(\mathcal{E}_S(\tilde{W})\right)^2\right\}.$$

We can get the following lower bound on $|A|$:

$$|A|\mathcal{E}_S(\tilde{W})^2 \geq \frac{1}{1.02^2}\sum_{j\in A}\|g_j\|_2^2$$

$$\geq \frac{1}{1.05}\left(\frac{1}{67}m_{L+1}\gamma^2[\mathcal{E}_S(\tilde{W})]^2 - \frac{1}{2\cdot 67}|A^c|\gamma^2[\mathcal{E}_S(\tilde{W})]^2\right)$$

$$\geq \frac{1}{1.05\cdot 2\cdot 67}m_{L+1}\gamma^2[\mathcal{E}_S(\tilde{W})]^2.$$

The first line follows by (28), and the second by writing the sum over $[m_{L+1}]$ as a sum over $A$ and $A^c$ and then (27) and the definition of $A$. The last line holds since $|A^c| \leq m_{L+1}$, and all of the above allows for the bound

$$|A| \geq \frac{1}{141}m_{L+1}\gamma^2. \tag{29}$$

Let now $A' = \{j \in [m_{L+1}] : \sigma'(\tilde{w}_{L+1,j}^\top\tilde{x}_{L,i}) \neq \sigma'(w_{L+1,j}^\top x_{L,i})\}$. By Lemma B.4, we have

$$|A'| = \left\|\tilde{\Sigma}_{L+1}(x) - \Sigma_{L+1}(x)\right\|_0 \leq C_1\tau^{\frac{2}{3}}m_{L+1}. \tag{30}$$

Since $\tau \leq \nu\gamma^3$, we can make $\nu$ small enough so that $C_1\tau^{\frac{2}{3}} < \gamma^2\cdot(1/141 - 1/150)$. Thus (29) and (30) imply

$$|A \setminus A'| \geq |A| - |A'| \geq \frac{1}{141}m_{L+1}\gamma^2 - C_1\tau^{\frac{2}{3}}m_{L+1} \geq \frac{1}{150}m_{L+1}\gamma^2. \tag{31}$$

By definition, $\nabla_{W_{L+1,j}}L_S(\tilde{W}) = \frac{1}{n}\sum_{i=1}^{n}\ell'(y_i\tilde{y}_i)\cdot v_j\cdot y_i\cdot\sigma'(\tilde{w}_{L+1,j}^\top\tilde{x}_{L,i})\cdot\tilde{x}_{L,i}$. For indices $j \in A \setminus A'$, we can therefore write

$$\|g_j\|_2 - \left\|\nabla_{W_{L+1,j}}L_S(\tilde{W})\right\|_2 \leq \left\|\frac{1}{n}\sum_{i=1}^{n}\ell'(y_i\tilde{y}_i)\cdot v_j\cdot y_i\cdot\sigma'(w_{L+1,j}^\top x_{L,i})\cdot(x_{L,i} - \tilde{x}_{L,i})\right\|_2$$

$$\leq \frac{1}{n}\sum_{i=1}^{n}\left\|\ell'(y_i\tilde{y}_i)\cdot v_j\cdot y_i\cdot\sigma'(w_{L+1,j}^\top x_{L,i})\cdot(x_{L,i} - \tilde{x}_{L,i})\right\|_2$$

$$\leq C_3\tau\mathcal{E}_S(\tilde{W}). \tag{32}$$

The first inequality follows by the triangle inequality and since indices $j \in A \setminus A'$ satisfy $\sigma'(\tilde{w}_{L+1,j}^\top\tilde{x}_{L,i}) = \sigma(w_{L+1,j}^\top x_{L,i})$. The second inequality is an application of Jensen inequality. The last inequality follows by Lemma B.4 and since $v_j, y_i \in \{\pm 1\}$. Now take $\nu$ small enough so that

$C_3 \tau < \left( (2 \cdot 67)^{-1/2} - 1/16 \right)$. Then we can use (32) together with the definition of $A$ to get for any index $j \in A \setminus A'$,

$$\left\| \nabla_{W_{L+1,j}} L_S(\tilde{W}) \right\|_2 \geq \frac{1}{\sqrt{2 \cdot 67}} \gamma \mathcal{E}_S(\tilde{W}) - C_3 \tau \mathcal{E}_S(\tilde{W}) \geq \frac{1}{16} \gamma \mathcal{E}_S(\tilde{W}). \tag{33}$$

Thus we can derive the lower bound for the gradient of the loss at the last layer:

$$\begin{aligned}
\left\| \nabla_{W_{L+1}} L_S(\tilde{W}) \right\|_F^2 &= \sum_{j=1}^{m_{L+1}} \left\| \nabla_{W_{L+1,j}} L_S(\tilde{W}) \right\|_F^2 \\
&\geq \sum_{j \in A \setminus A'} \left\| \nabla_{W_{L+1,j}} L_S(\tilde{W}) \right\|_2^2 \\
&\geq \frac{1}{16^2} |A \setminus A'| \gamma^2 [\mathcal{E}_S(\tilde{W})]^2 \\
&\geq \frac{1}{150 \cdot 16^2} \gamma^4 m_{L+1} [\mathcal{E}_S(\tilde{W})]^2.
\end{aligned}$$

The first line is by definition, and the second is since the spectral norm is at most the Frobenius norm. The third line uses (33), and the final inequality comes from (31). $\qquad\square$

## B.4 Proof of Lemma 4.4: gradient upper bound

*Proof.* Using the gradient formula (2) and the $H_l^{l'}$ notation from (1), we can write

$$\nabla_{W_l} L_S(\tilde{W}) = \theta^{\mathbb{1}(2 \leq l \leq L)} \frac{1}{n} \sum_{i=1}^n \ell'(y_i \tilde{y}_i) \cdot y_i \cdot \tilde{x}_{l-1,i} v^\top \tilde{H}_{l+1}^{L+1} \tilde{\Sigma}_l(x_i), \quad (1 \leq l \leq L+1). \tag{34}$$

Since $\tau \leq 1$, there is a constant $C$ such that w.h.p. $\left\| \tilde{W}_l \right\|_2 \leq C$ for all $l$. Thus, it is easy to see that an analogous version of Lemma B.2 can be applied with Lemma B.4 to get that with probability at least $1 - \delta$, for all $i \in [n]$ and for all $l$,

$$\|\tilde{x}_{l-1,i}\|_2 \leq C_1 \qquad \text{and} \qquad \left\| \tilde{H}_{l+1}^{L+1} \right\|_2 \leq C_2. \tag{35}$$

Therefore, we can bound

$$\begin{aligned}
\left\| \nabla_{W_l} L_S(\tilde{W}) \right\|_F &\leq \frac{1}{n} \sum_{i=1}^n \left\| \ell'(y_i \tilde{y}_i) \cdot y_i \cdot \tilde{x}_{l-1,i} v^\top \tilde{H}_{l+1}^{L+1} \tilde{\Sigma}_{l+1}(x_i) \right\|_F \\
&= \frac{1}{n} \sum_{i=1}^n \left\| \ell'(y_i \tilde{y}_i) \cdot y_i \cdot \tilde{x}_{l-1,i} \right\|_2 \left\| v^\top \tilde{H}_{l+1}^{L+1} \tilde{\Sigma}_{l+1}(x_i) \right\|_2 \\
&\leq C_3 \sqrt{m} \mathcal{E}_S(\tilde{W}).
\end{aligned}$$

The first line follows by the triangle inequality, and the second since for vectors $a, b$, we have $\left\| ab^\top \right\|_F = \|a\|_2 \|b\|_2$. The last line is by Cauchy–Schwarz, (35), and the definition of $\mathcal{E}_S$, finishing the case $l = 1$. By substituting the definition of the gradient of the loss using the formula (34) we may similarly demonstrate the corresponding bounds for $l \geq 2$ with an application of Cauchy–Schwartz. $\qquad\square$

## C  Proofs of Technical Lemmas

In this section we go over the proofs of the technical lemmas that were introduced in Appendix B. In the course of proving these technical lemmas, we will need to introduce a handful of auxiliary lemmas, whose proofs we leave for Appendix D. Throughout this section, we continue to assume that $\theta = 1/\Omega(L)$.

## C.1 Proof of Lemma B.2: intermediate layers are bounded

By Lemma B.1, there is a constant $C_1$ such that with probability at least $1 - \delta$, $\|W_l\|_2 \leq C_1$ for all $l = a, \ldots, b$. Therefore for each $r \geq 2$, we have

$$\left\| I + \theta \tilde{\Sigma}_r W_r \right\|_2 \leq \|I\|_2 + \theta \left\| \tilde{\Sigma}_r \right\|_2 \|W_r\|_2 \leq 1 + \theta C_1.$$

The submultiplicative property of the spectral norm gives

$$\left\| (I + \theta \tilde{\Sigma}_b W_b^\top)(I + \theta \tilde{\Sigma}_{b-1} W_{b-1}^\top) \cdot \ldots \cdot (I + \theta \tilde{\Sigma}_a W_a^\top) \right\|_2$$

$$\leq \prod_{r=a}^{b} \left\| I + \theta \tilde{\Sigma}_r W_r^\top \right\|_2$$

$$\leq (1 + \theta C_1)^L$$

$$\leq \exp\left( C_1 \theta L \right).$$

The result follows by the choice of scale $\theta = 1/\Omega(L)$ and taking $\theta$ small.

## C.2 Proof of Lemma B.3: Lipschitz property with respect to input space at each layer

Before beginning with the proof, we introduce the following claim that will allow us to develop a Lipschitz property with respect to the weights. This was used in Cao and Gu [5] and Allen-Zhu et al. [1].

**Claim C.1.** For arbitrary $u, y \in \mathbb{R}^{m_l}$, let $D(u)$ be the diagonal matrix with diagonal entries $[D(u)]_{j,j} = \mathbb{1}(u_j \geq 0)$. Then there exists another diagonal matrix $\check{D}(u)$ such that $\left\| D(u) + \check{D}(u) \right\|_2 \vee \left\| \check{D}(u) \right\|_2 \leq 1$ and $\sigma(u) - \sigma(y) = \left( D(u) + \check{D}(u) \right)(u - y)$.

*Proof of Claim C.1.* Simply define

$$[\check{D}(u)]_{j,j} = \begin{cases} [D(u) - D(y)] \frac{y_j}{u_j - y_j} & u_j \neq y_j, \\ 0 & u_j = y_j. \end{cases}$$

$\square$

*Proof of Lemma B.3.* We note that for any $x, y$, the matrix $|\Sigma_l(x) - \Sigma_l(y)|$ is zero everywhere except possibly the diagonal where it is either zero or one. Therefore its spectral norm is uniformly bounded by 1 for all $x, y$. Using this, Lemma B.1 gives with probability at least $1 - \delta/3$, for all $x, x' \in S^{d-1}$,

$$\|x_1 - x_1'\|_2 = \left\| (\Sigma_1(x_1) - \Sigma_1(x_1')) W_1^\top (x - x') \right\|_2$$

$$\leq \|\Sigma_1(x_1) - \Sigma_1(x_1')\|_2 \|W_1\|_2 \|x - x'\|_2$$

$$\leq 1 \cdot C \cdot \|x - x'\|_2.$$

For the case $L \geq l \geq 2$, we have residual links to analyze. Using Claim C.1 we can write

$$\sigma(W_l^\top x_{l-1}) - \sigma(W_l^\top \widehat{x}_{l-1}) = (\Sigma_l(x) + \check{\Sigma}_l(x)) W_l^\top (x_{l-1} - \widehat{x}_{l-1})$$

for diagonal matrix $\check{\Sigma}_l$ satisfying $\left\| \check{\Sigma}_l(x) \right\|_2 \leq 1$ and $\left\| \Sigma_l(x) + \check{\Sigma}_l(x) \right\|_2 \leq 1$. By Lemma B.2, we have with probability at least $1 - \delta/3$, for all $2 \leq l \leq L$ and all $x, x' \in S^{d-1}$,

$$\|x_l - x_l'\|_2 \leq \left\| I + \theta(\Sigma_l(x) + \check{\Sigma}_l(x)) W_l^\top \right\|_2 \|x_{l-1} - x_{l-1}'\|_2$$

$$\leq (1 + \theta C_0) \|x_{l-1} - x_{l-1}'\|_2$$

$$\leq \left( 1 + \frac{C_0 \theta L}{L} \right)^L \cdot \|x - x'\|_2$$

$$\leq C_1 \|x - x'\|_2,$$

since $\theta L$ is uniformly bounded from above.

The case $l = L + 1$ follows as in the case $l = 1$ by an application of Lemma B.1, so that with probability at least $1 - \delta/3$, $\left\| x'_{L+1} - x_{L+1} \right\|_2 \leq C_2 \left\| x - x' \right\|_2$. Putting the above three claims together, we get a constant $C_3$ such that with probability at least $1 - \delta$, $\left\| x_l - x'_l \right\|_2 \leq C_3 \left\| x - x' \right\|_2$ for all $x, x' \in \mathcal{S}^{d-1}$ and for all $l \in [L + 1]$.

$\square$

### C.3  Proof of Lemma B.4: local Lipschitz property with respect to weights and sparsity bound

For this lemma, we need to introduce an auxiliary lemma that allows us to get control over the sparsity levels of the ReLU activation patterns. Its proof can be found in Appendix D.1.

**Lemma C.2.** There are absolute constants $C, C'$ such that for any $\delta > 0$, if

$$m \geq C \left( \beta^{-1} \sqrt{d \log \frac{1}{\beta \delta}} \vee d \log \frac{mL}{\delta} \right),$$

then with probability at least $1 - \delta$, the sets

$$\mathcal{S}_l(x, \beta) = \{ j \in [m_l] : |w_{l,j}^\top x_{l-1}| \leq \beta \}, \; x \in S^{d-1}, \; l \in [L + 1],$$

satisfy $|\mathcal{S}_l(\beta)| \leq C' m_l^{\frac{3}{2}} \beta$ for all $x \in S^{d-1}$ and $l \in [L + 1]$.

*Proof of Lemma B.4.* We begin with the Lipschitz property, and afterwards will show the sparsity bound. Consider $l = 1$. Since $\widehat{x}_1 = \sigma \left( \widehat{W}_1^\top x \right)$ and $\tilde{x}_1 = \sigma \left( \tilde{W}_1^\top x \right)$, by Claim C.1, for every $l$ there is a diagonal matrix $\check{\Sigma}_l(x)$ with $\left\| \check{\Sigma}_l(x) \right\|_2 \leq 1$ and $\left\| \widehat{\Sigma}_l(x) + \check{\Sigma}_l(x) \right\|_2 \leq 1$ such that

$$
\begin{aligned}
\left\| \widehat{x}_1 - \tilde{x}_1 \right\|_2 &= \left\| \left( \widehat{\Sigma}_1(x) + \check{\Sigma}_1(x) \right) \left( \widehat{W}_1^\top x - \tilde{W}_1^\top x \right) \right\|_2 \\
&\leq \left\| \widehat{\Sigma}_1(x) + \check{\Sigma}_1(x) \right\|_2 \left\| \widehat{W}_1 - \tilde{W}_1 \right\|_2 \left\| x \right\|_2 \\
&\leq \left\| \widehat{W}_1 - \tilde{W}_1 \right\|_2 .
\end{aligned}
\tag{36}
$$

For $l = 2, \ldots, L$, we can write

$$
\begin{aligned}
\widehat{x}_l - \tilde{x}_l &= \widehat{x}_{l-1} + \theta \sigma \left( \widehat{W}_l^\top \widehat{x}_{l-1} \right) - \tilde{x}_{l-1} - \theta \sigma \left( \tilde{W}_l^\top \tilde{x}_{l-1} \right) \\
&= \left[ I + \theta \left( \widehat{\Sigma}_l(x) + \check{\Sigma}_l(x) \right) \tilde{W}_l^\top \right] (\widehat{x}_{l-1} - \tilde{x}_{l-1}) + \theta \left[ \widehat{\Sigma}_l(x) + \check{\Sigma}_l(x) \right] \left( \widehat{W}_l - \tilde{W}_l \right)^\top \widehat{x}_{l-1}.
\end{aligned}
$$

Therefore, we have

$$
\begin{aligned}
\left\| \widehat{x}_l - \tilde{x}_l \right\|_2 &\leq \left\| I + \theta (\widehat{\Sigma}_l(x) + \check{\Sigma}_l(x)) \tilde{W}_l^\top \right\|_2 \left\| \widehat{x}_{l-1} - \tilde{x}_{l-1} \right\|_2 + \theta \left\| \widehat{\Sigma}_l(x) + \check{\Sigma}_l(x) \right\|_2 \left\| \widehat{W}_l - \tilde{W}_l \right\|_2 \left\| \widehat{x}_{l-1} \right\|_2 \\
&\leq (1 + C\theta) \left\| \widehat{x}_{l-1} - \tilde{x}_{l-1} \right\|_2 + \theta \left\| \widehat{W}_l - \tilde{W}_l \right\|_2 \left\| \widehat{x}_{l-1} \right\|_2 .
\end{aligned}
\tag{37}
$$

We notice an easy induction will complete the proof. For the base case $l = 2$, notice that $\left\| \widehat{x}_1 \right\|_2 \leq \left\| x_1 \right\|_2 + \left\| \widehat{x}_1 - x_1 \right\|_2 \leq C + \tau \leq C'$, so that (36) and (37) give

$$\left\| \widehat{x}_2 - x_2 \right\|_2 \leq (1 + C\theta) \left\| \widehat{W}_1 - \tilde{W}_1 \right\|_2 + C' \theta \left\| \widehat{W}_2 - \tilde{W}_2 \right\|_2 \leq C_4 \left\| \widehat{W}_1 - \tilde{W}_1 \right\|_2 + C_4 \theta \left\| \widehat{W}_2 - \tilde{W}_2 \right\|_2 .$$

Suppose by induction that there exists a constant $C$ such that $\left\| \widehat{x}_{l-1} - x_{l-1} \right\|_2 \leq C_5 \left\| \widehat{W}_1 - \tilde{W}_1 \right\|_2 + C_5 \theta \sum_{r=1}^{l-1} \left\| \widehat{W}_r - \tilde{W}_r \right\|_2$. Then as in the base case, $\left\| \widehat{x}_{l-1} \right\|_2 \leq C'$, so that (37) gives for all $l = 2, \ldots, L$,

$$
\begin{aligned}
\left\| \widehat{x}_l - \tilde{x}_l \right\|_2 &\leq (1 + C\theta) C \left[ C_5 \left\| \widehat{W}_1 - \tilde{W}_1 \right\|_2 + C_5 \theta \sum_{r=1}^{l-1} \left\| \widehat{W}_r - \tilde{W}_r \right\|_2 \right] + C' \theta \left\| \widehat{W}_l - \tilde{W}_l \right\|_2 \\
&\leq C_6 \left\| \widehat{W}_1 - \tilde{W}_1 \right\|_2 + C_6 \theta \sum_{r=1}^{l} \left\| \widehat{W}_r - \tilde{W}_r \right\|_2 .
\end{aligned}
$$

Finally, the case $l = L + 1$ follows similarly to the case $l \leq L$, as

$$\|\widehat{x}_{L+1} - \tilde{x}_{L+1}\|_2 = \left\| \left( \widehat{\Sigma}_{L+1}(x) + \check{\Sigma}_{L+1}(x) \right) \left( \widehat{W}_{L+1}^\top \widehat{x}_L - \tilde{W}_{L+1}^\top \tilde{x}_L \right) \right\|_2$$

$$\leq C \left\| \widehat{W}_{L+1} - \tilde{W}_{L+1} \right\|_2 + C' \|\widehat{x}_L - \tilde{x}_L\|_2 .$$

The bound for the sparsity levels of $\tilde{\Sigma}_l(x) - \widehat{\Sigma}_l(x)$ follows the same proof as Lemma B.5 in Cao and Gu [5] with an application of our Lemma C.2. Sketching this proof, we note that it suffices to prove a bound for $\left\| \widehat{\Sigma}_l(x) - \Sigma_l(x) \right\|_0$, use the same proof for $\left\| \tilde{\Sigma}_l(x) - \Sigma_l(x) \right\|_0$ and then use triangle inequality to get the final result. We write

$$\left\| \widehat{\Sigma}_l(x) - \Sigma_l(x) \right\|_0 = s_l^{(1)}(\beta) + s_l^{(2)}(\beta),$$

where

$$s_l^{(1)}(\beta) = |\{j \in \mathcal{S}_l(x, \beta) : (\widehat{w}_{l,j}^\top \widehat{x}_{l-1}) \cdot (w_{l,j}^\top x_{l-1}) < 0\}|,$$

$$s_l^{(2)}(\beta) = |\{j \in \mathcal{S}_l^c(x, \beta) : (\widehat{w}_{l,j}^\top \widehat{x}_{l-1}) \cdot (w_{l,j}^\top x_{l-1}) < 0\}|,$$

which leads to

$$\left\| \widehat{\Sigma}_l(x) - \Sigma_l(x) \right\|_0 \leq C m^{\frac{3}{2}} \beta + C_5 \tau^2 \beta^{-2}.$$

The choice of $\beta = m_l^{-\frac{1}{2}} \tau^{\frac{2}{3}}$ completes the proof. $\qquad \square$

## C.4 Proof of Lemma B.5: behavior of network output in $\mathcal{W}(W^{(0)}, \tau)$ when acting on sparse vectors

This technical lemma will require two auxiliary lemmas before we may begin the proof. Their proofs are left for Appendix D.2 and D.3.

**Lemma C.3.** Consider the function $g_l : \mathbb{R}^{m_l} \times \mathbb{R}^{m_{L+1}} \to \mathbb{R}$ defined by

$$g_l(a, b) := b^\top W_{L+1}^\top \xi_l a, .$$

where $\xi_l \in \mathbb{R}^{m_L \times m_l}$, and $l \geq 2$. Suppose that with probability at least $1 - \delta/2$, $\|\xi_l\|_2 \leq C$ holds for all $\xi_l$, $l = 2, \ldots, L$. If $s \log m = \Omega \left( C \log(L/\delta) \right)$, then there is a constant $C_0 > 0$ such that probability at least $1 - \delta$, for all $l$,

$$\sup_{\|a\|_2 = \|b\|_2 = 1, \ \|a\|_0, \|b\|_0 \leq s} |g_l(a, b)| \leq C_0 \sqrt{\frac{1}{m} s \log m}.$$

**Lemma C.4.** Consider the function $g_l : \mathbb{R}^{m_l} \to \mathbb{R}$ defined by

$$g_l(a) := v^\top \Sigma_{L+1}(x)^\top W_{L+1}^\top \xi_l a,$$

where $\xi_l \in \mathbb{R}^{m_L \times m_l}$ and $l \geq 2$. Assume that with probability at least $1 - \delta$, $\|\xi_l\|_2 \leq C_0$ for all $l$. Then provided $s \log m = \Omega \left( \log(L/\delta) \right)$, we have with probability at least $1 - \delta$, for all $l$,

$$\sup_{\|a\|_2 = 1, \ \|a\|_0 \leq s} |g_l(a)| \leq C_1 \sqrt{s \log m}.$$

With these lemmas in place, we can prove Lemma B.5.

*Proof of Lemma B.5.* By definition, $g_l(a, x) = v^\top \tilde{H}_l^{L+1} a$. First: since $\left\| \tilde{W}_l - W_l \right\|_2 \leq \tau$, there is an absolute constant $C_2 > 0$ such that with high probability, $\left\| \tilde{W}_l \right\|_2 \leq C_2$ for all $l$. Therefore, we have with high probability for all $x \in S^{d-1}$, all $l$, and all $a$ considered,

$$\left\| \tilde{H}_l^L \right\|_2 \leq \left[ \prod_{r=l}^{L} \left\| I + \theta \tilde{\Sigma}_r(x) \tilde{W}_r^\top \right\|_2 \right] \|a\|_2 \leq (1 + \theta \cdot 1 \cdot C_2)^L \cdot 1 \leq C_3, \tag{38}$$

by our choice of $\theta$. We proceed by bounding $g_l$ by a sum of four terms:

$$|g_l(a,x)| \leq a \leq \left| v^\top \left( \tilde{\Sigma}_{L+1}(x) - \Sigma_{L+1}(x) \right) \tilde{W}_{L+1}^\top \tilde{H}_l^L a \right| + \left| v^\top \Sigma_{L+1}(x) \check{W}_{L+1}^\top \tilde{H}_l^L a \right|$$

$$\leq \left| v^\top \left( \tilde{\Sigma}_{L+1}(x) - \Sigma_{L+1}(x) \right) \left( \tilde{W}_{L+1}^\top - W_{L+1}^\top \right) \tilde{H}_l^L a \right| + \left| v^\top \left( \tilde{\Sigma}_{L+1}(x) - \Sigma_{L+1}(x) \right) W_{L+1}^\top \tilde{H}_l^L a \right|$$

$$+ \left| v^\top \Sigma_{L+1}(x) \left( \tilde{W}_{L+1}^\top - W_{L+1}^\top \right) \tilde{H}_l^L a \right| + \left| v^\top \Sigma_{L+1}(x) W_{L+1}^\top \tilde{H}_l^L a \right|.$$

For the first term, we can write

$$\left| v^\top \left( \tilde{\Sigma}_{L+1}(x) - \Sigma_{L+1}(x) \right) \left( \tilde{W}_{L+1}^\top - W_{L+1}^\top \right) \tilde{H}_l^L \right|$$

$$\leq \|v\|_2 \left\| \left( \tilde{\Sigma}_{L+1}(x) - \Sigma_{L+1}(x) \right) \left( \tilde{W}_{L+1}^\top - W_{L+1}^\top \right) H_l^L a \right\|_2$$

$$\leq C\sqrt{m} \left\| \tilde{\Sigma}_{L+1}(x) - \Sigma_{L+1}(x) \right\|_2 \left\| \tilde{W}_{L+1} - W_{L+1} \right\|_2 \left\| \tilde{H}_l^L a \right\|_2$$

$$\leq C'\tau\sqrt{m},$$

where we have used Cauchy–Schwarz in the first line, properties of the spectral norm in the second, and (38) in the third. A similar calculation shows

$$\left| v^\top \Sigma_{L+1} \left( \tilde{W}_{l+1}^\top - W_{L+1}^\top \right) \tilde{H}_l^L \right| \leq \|v\|_2 \left\| \Sigma_{L+1} \left( \tilde{W}_{L+1}^\top - W_{L+1}^\top \right) \tilde{H}_l^L \right\|_2$$

$$\leq C\tau\sqrt{m}.$$

For the second and fourth terms, we use Lemmas C.3 and C.4. Let $\check{b}^\top = v^\top \left( \tilde{\Sigma}_{L+1}(x) - \Sigma_{L+1}(x) \right)$. Then it is clear that $\|\check{b}\|_0 \leq s$ and $\|\check{b}\|_2 \leq \sqrt{m}$ (in fact, $\|\check{b}\|_2 \leq \sqrt{s}$, but this doesn't matter since the fourth term dominates the second term). Thus applying Lemma C.3 to $b = \check{b}/\|\check{b}\|_2$,

$$|v^\top \left( \tilde{\Sigma}_{L+1}(x) - \Sigma_{L+1}(x) \right) W_{L+1}^\top \tilde{H}_l^L a| \leq C\sqrt{m} \cdot \sqrt{\frac{s}{m} \log m}$$

$$\leq C\sqrt{s \log m}.$$

For the fourth term, we can directly apply Lemma C.4 to get another term $\propto \sqrt{s \log m}$. $\qquad\square$

## C.5 Proof of Lemma B.6

This lemma is the key to the sublinear dependence on $L$ for the required width for the generalization result. Essential to its proof is the following proposition which states that there is a linear separability condition at each layer due to Assumption 3.2 with only a logarithmic dependence on the depth $L$. In fact, we only need linear separability at the second-to-last layer for the proof of Lemma B.6.

**Proposition C.5.** Suppose $m \geq C\gamma^{-2} \left( d\log\frac{1}{\gamma} + \log\frac{L}{\delta} \right)$ for some large constant $C$. Then there exists $\alpha \in S^{m_L-1}$ such that with probability at least $1 - \delta$, for all $l = 1, \ldots, L$, we have

$$y \langle \alpha, x_l \rangle \geq \gamma/2.$$

*Proof of Proposition C.5.* We recall that Assumption 3.2 implies that there exists $c(\bar{u})$ with $\|c(u)\|_\infty \leq 1$ such that $f(x) = \int_{\mathbb{R}^d} c(u)\sigma(u^\top x)p(u)du$ satisfies $y \cdot f(x) \geq \gamma$ for all $(x,y) \in \text{supp}(\mathcal{D})$. Following Lemma C.1 in Cao and Gu [5], if we define

$$\alpha := \sqrt{\frac{1}{m_1}} \cdot \left( c\left( \sqrt{\frac{m_1}{2}} w_{1,1} \right), \ldots, c\left( \sqrt{\frac{m_1}{2}} w_{1,m_1} \right) \right),$$

then $\alpha = \alpha'/\|\alpha'\|_2 \in S^{m_1-1}$ satisfies $y \cdot \alpha^\top x_1 \geq \frac{\gamma}{2}$ for all $(x,y) \in \text{supp}\,\mathcal{D}$.

We now show that the $l$-th layer activations $x_l$ are linearly separable using $\alpha$. We can write, for $l = 2, \ldots, L$,

$$\langle \alpha, x_l \rangle = \langle \alpha, (I + \theta\Sigma_l(x)W_l^\top)x_{l-1} \rangle$$

$$= \langle \alpha, x_1 \rangle + \theta \sum_{l'=2}^{l} \langle \alpha, \Sigma_{l'}(x)W_{l'}^\top x_{l'-1} \rangle. \tag{39}$$

Since $\langle \alpha, \Sigma_l(x) W_l^\top x_{l-1} \rangle = \sum_{k=1}^{m_l} \sqrt{\frac{1}{m_1}} c \left( \sqrt{\frac{m_1}{2}} w_{1,k} \right) \cdot \sigma(w_{l,k}^\top x_{l-1})$ and $\|c(\cdot)\|_\infty \leq 1$, we have for every $l \geq 2$,

$$-\sum_{k=1}^{m_l} \sqrt{\frac{1}{m_1}} \left| w_{l,k}^\top x_{l-1} \right| \leq \langle \alpha, \Sigma_l(x) W_l^\top x_{l-1} \rangle \leq \sum_{k=1}^{m_l} \sqrt{\frac{1}{m_1}} \left| w_{l,k}^\top x_{l-1} \right|. \qquad (40)$$

Thus it suffices to find an upper bound for the term on the r.h.s. of (40). Since we have

$$\mathbb{E} \left| w_{l,k}^\top x_{l-1} \right| = \sqrt{\frac{2}{\pi}} \sqrt{\frac{2}{m_1}} \|x_{l-1}\|_2 \leq C_2 m^{-\frac{1}{2}},$$

we can apply Hoeffding inequality to get absolute constants $C_4, C_5 > 0$ such that for fixed $x$ and $l$, we have with probability at least $1 - \delta$,

$$\sum_{k=1}^{m_l} \sqrt{\frac{1}{m_1}} \left| w_{l,k}^\top x_{l-1} \right| \leq \sum_{k=1}^{m_l} \sqrt{\frac{1}{m}} C_2 m^{-\frac{1}{2}} + C_4 \sqrt{\frac{1}{m} \log \frac{1}{\delta}}$$

$$\leq C_5 + C_4 \sqrt{\frac{1}{m} \log \frac{1}{\delta}}.$$

Take a $\frac{1}{2}$-net $\mathcal{N}$ of $S^{d-1}$ so that $|\mathcal{N}| \leq 5^d$ and every $x \in S^{d-1}$ has $\widehat{x} \in \mathcal{N}$ with $\|x - \widehat{x}\|_2 \leq \frac{1}{2}$. Then, provided $m \geq Cd \log \frac{L}{\delta}$, there is a constant $C_6 > 0$ such that we have with probability at least $1 - \delta$, for all $\widehat{x} \in \mathcal{N}$ and all $l \leq L$,

$$\sum_{k=1}^{m_l} \sqrt{\frac{1}{m_1}} \left| w_{l,k}^\top \widehat{x}_{l-1} \right| \leq C_6.$$

By (40), this means for all $\widehat{x} \in \mathcal{N}$ and $l$, $-C_6 \leq \langle \alpha, \Sigma_l(\widehat{x}) W_l^\top \widehat{x}_{l-1} \rangle \leq C_6$. We can lift this to hold over $S^{d-1}$ by using Lemma B.3: for arbitrary $x \in S^{d-1}$ we have

$$\left| \langle \alpha, \Sigma_l(x) W_l^\top x_l \rangle \right| \leq \left| \langle \alpha, \Sigma_l(x) W_l^\top (x_l - \widehat{x}_l) \rangle \right| + \left| \langle \alpha, \Sigma_l(x) W_l^\top \widehat{x}_l \rangle \right|$$

$$\leq \|\tilde{\alpha}_l\|_2 \|\Sigma_l(x)\|_2 \|W_l\|_2 \|x_l - \widehat{x}_l\|_2 + C_6$$

$$\leq C_7,$$

so that with probability at least $1 - \delta$, for all $l \leq L$ and all $x \in S^{d-1}$, we have

$$-C_7 \leq \langle \alpha, \Sigma_l(x) W_l^\top \widehat{x}_{l-1} \rangle \leq C_7.$$

Substituting the above into (39), we get

$$\begin{cases} \langle \alpha, x_l \rangle \geq \langle \alpha, x_1 \rangle - \theta L C_7, \\ -\langle \alpha, x_l \rangle \geq -\langle \alpha, x_1 \rangle - \theta L C_7. \end{cases}$$

Considering the cases $y = \pm 1$ we thus get with probability at least $1 - \delta$ for all $l$ and $(x, y) \in \operatorname{supp} \mathcal{D}$,

$$\begin{cases} y \langle \alpha, x_l \rangle \geq y \langle \alpha, x_1 \rangle - \theta L C_7 \geq \frac{\gamma}{2} - \theta L C_7, & y = 1, \\ y \langle \alpha, x_l \rangle \geq y \langle \alpha, x_1 \rangle - \theta L C_7 \geq \frac{\gamma}{2} - \theta L C_7, & y = -1. \end{cases}$$

Thus taking $\theta$ small enough so that $\theta L \leq \gamma C_7^{-1}/4$ completes the proof. $\qquad \square$

With Proposition C.5 in hand, we can prove Lemma B.6.

*Proof of Lemma B.6.* By Proposition C.5, there exists $\alpha_L \in S^{m_L - 1}$ such that with probability at least $1 - \delta$, $y \langle \alpha_L, x_L \rangle \geq \gamma/4$ for all $(x, y) \in \operatorname{supp}(\mathcal{D})$. In particular, since $a$ is non-negative, this implies for all $i$,

$$\langle a(x_i, y_i) \cdot y_i \cdot x_{L,i}, \alpha_L \rangle = a(x_i, y_i) \cdot y_i \langle x_{L,i}, \alpha_L \rangle \geq a(x_i, y_i) y_i \gamma/4. \qquad (41)$$

Since $\mathbb{E}[\sigma'(w_{L+1,j}^\top x_{L,i}) | x_{L,i}] = \frac{1}{2}$, by Hoeffding inequality, with probability at least $1 - \delta/2$, for all $i = 1, \ldots, n$, we have

$$\frac{1}{m_{L+1}} \sum_{j=1}^{m_{L+1}} \sigma'(w_{L+1,j}^\top x_{L,i}) \geq \frac{1}{2} - C_1 \sqrt{\frac{1}{m_{L+1}} \log(n/\delta)} \geq \frac{49}{100}. \qquad (42)$$

Therefore, we can bound

$$\sum_{j=1}^{m_{L+1}} \left\| \frac{1}{n} \sum_{i=1}^{n} \left[ a(x_i, y_i) \cdot y_i \cdot \sigma'(w_{L+1,j}^\top x_{L,i}) \cdot x_{L,i} \right] \right\|_2^2$$

$$\geq m_{L+1} \left\| \frac{1}{m_{L+1}} \sum_{j=1}^{m_{L+1}} \frac{1}{n} \sum_{i=1}^{n} \left[ a(x_i, y_i) \cdot y_i \cdot \sigma'(w_{L+1,j}^\top x_{L,i}) \cdot x_{L,i} \right] \right\|_2^2$$

$$= m_{L+1} \left\| \frac{1}{n} \sum_{i=1}^{n} \left[ a(x_i, y_i) \cdot y_i \cdot x_{L,i} \frac{1}{m_{L+1}} \sum_{j=1}^{m_{L+1}} \sigma'(w_{L+1,j}^\top x_{L,i}) \right] \right\|_2^2$$

$$\geq m_{L+1} \left\langle \frac{1}{n} \sum_{i=1}^{n} a(x_i, y_i) \cdot y_i \cdot x_{L,i} \cdot \frac{1}{m_{L+1}} \sum_{j=1}^{m_{L+1}} \sigma'(w_{L+1,j}^\top x_{L,i}), \alpha_L \right\rangle^2$$

$$= m_{L+1} \left( \frac{1}{n} \sum_{i=1}^{n} a(x_i, y_i) \cdot y_i \cdot \frac{1}{m_{L+1}} \sum_{j=1}^{m_{L+1}} \sigma'(w_{L+1,j}^\top x_{L,i}) \cdot \langle x_{L,i}, \alpha_L \rangle \right)^2$$

$$\geq \left( \frac{49}{100} \right)^2 m_{L+1} \left( \frac{1}{n} \sum_{i=1}^{n} a(x_i, y_i) \right)^2 \cdot \frac{\gamma^2}{4^2}$$

$$\geq \frac{1}{67} m_{L+1} \cdot \gamma^2 \left( \frac{1}{n} \sum_{i=1}^{n} a(x_i, y_i) \right)^2.$$

The first inequality above follows by Jensen inequality. The second inequality follows by Cauchy–Schwarz and since $\|\alpha_L\|_2 = 1$. The third inequality follows with an application of (41) and (42), and the final inequality by arithmetic. $\square$

# D  Proofs of Auxiliary Lemmas

## D.1  Proof of Lemma C.2

*Proof.* By following a proof similar to that of Lemma A.8 in Cao and Gu [5], one can easily prove the following claim:

**Claim D.1.** For $v \in \mathbb{R}^{m_{l-1}}$, $\beta > 0$, and $l \in [L+1]$ define

$$\mathcal{S}_l(v, \beta) := \{ j \in [m_l] : |w_{l,j}^\top v| \leq \beta \}. \tag{43}$$

Suppose that there is an absolute constant $\xi \in (0,1)$ such that for any $\delta > 0$ we have with probability at least $1 - \delta/2$, $\|v\|_2 \geq \xi$ for all $v \in \mathcal{V}$ for some finite set $\mathcal{V} \subset \mathbb{R}^{m_{l-1}}$. Then there exist absolute constants $C, C' > 0$ such that if $m \geq C\beta^{-1}\sqrt{\log(4|\mathcal{V}|/\delta)}$, then with probability at least $1 - \delta$, we have $|\mathcal{S}_l(v, \beta)| \leq C' m_l^{3/2} \beta$ for all $v \in \mathcal{V}$.

By Lemmas 4.1 and B.1, with probability at least $1 - \delta/3$, we have $\|x_{l-1}\|_2 \geq C$ and $\|w_{l,j}\|_2 \leq C_1$ for all $x \in S^{d-1}$, $l \in [L+1]$, and $j \in [m_l]$. By Lemma B.3, with probability at least $1 - \delta/3$, we have $\|x_l - x_l'\|_2 \leq C_2 \|x - x'\|_2$ for all $x, x' \in S^{d-1}$. By taking $\mathcal{V}$ to be the $\beta/(C_1 C_2)$-net $\mathcal{N}(S^{d-1}, \beta/(C_1 C_2))$, since $|\mathcal{N}| \leq (4 C_1 C_2/\beta)^d$, the assumption that $m \geq C\beta^{-1}\sqrt{d \log(1/(\beta\delta))}$ allows us to apply Lemma D.1 to get that with probability at least $1 - \delta/3$, we have $|\mathcal{S}_l(\widehat{x}, 2\beta)| \leq 2C' m_l^{\frac{3}{2}} \beta$ for all $l$ and $\widehat{x} \in \mathcal{N}$. For arbitrary $x \in S^{d-1}$, there exists $\widehat{x} \in \mathcal{N}$ with $\|x - \widehat{x}\|_2 \leq \beta/(C_1 C_2)$. Thus, we have

$$\begin{aligned}
|w_{l,j}^\top x_{l-1}| &\leq |w_{l,j}^\top \widehat{x}_{l-1}| + |w_{l,j}^\top (x_{l-1} - \widehat{x}_{l-1})| \\
&\leq \beta + \|w_{l,j}\|_2 \|x_{l-1} - \widehat{x}_{l-1}\|_2 \\
&\leq \beta + C_1 \cdot C_2 \|x - \widehat{x}\|_2 \\
&\leq 2\beta,
\end{aligned}$$

i.e. $\mathcal{S}_l(x, \beta) \subset \mathcal{S}_l(\widehat{x}, 2\beta)$. Therefore $|\mathcal{S}_l(x, \beta)| \leq |\mathcal{S}_l(\widehat{x}, 2\beta)| \leq 2C'm_l^{\frac{3}{2}}\beta$, as desired. $\qquad\square$

## D.2 Proof of Lemma C.3

*Proof.* The $j$-th row of $W_{L+1}^\top \xi_l a$ has distribution $w_{L+1,j}^\top \xi_l a \sim N\left(0, \frac{2}{m_{L+1}}\|\xi_l a\|_2^2\right)$, and hence $g_l(a, b) \sim N\left(0, \frac{2}{m_l}\|\xi_l a\|_2^2\right)$. Since $\|\xi_l\|_2 \leq C_0$ for all $l$ with high probability, it is clear that $\|\xi_l a\|_2^2 \leq C_0^2$. Thus applying Hoeffding inequality gives a constant $C_3 > 0$ such that we have for fixed $a$ and $b$, with probability at least $1 - \delta$,

$$|b^\top W_{L+1}^\top \xi_l a| \leq C_3\sqrt{\frac{1}{m_{L+1}}\log\frac{1}{\delta}}. \tag{44}$$

Let $\mathcal{M}_a$ be a fixed subspace of $\mathbb{R}^{m_l}$ with sparsity $s$, and let $\mathcal{N}_a(\mathcal{M}, 1/4)$ be a $1/4$-net covering $\mathcal{M}_a$. There are $\binom{m_l}{s}$ choices of such $\mathcal{M}_a$. Let $\mathcal{N}_a = \cup_{\mathcal{M}_a}\mathcal{N}_a(\mathcal{M}_a, 1/4)$ be the union of such spaces. By Lemma 5.2 in Vershynin [23], for $s$ larger than e.g. 15, we have

$$|\mathcal{N}_a| \leq \binom{m_l}{s}9^s \leq m_l^s.$$

Similarly consider subspace $\mathcal{M}_b \subset \mathbb{R}^{m_{L+1}}$ with sparsity level $s$ and let $\mathcal{N}_b(\mathcal{M}_b, 1/4)$ be a $1/4$-net of $\mathbb{R}^{m_{L+1}}$ with sparsity level $s$ and define $\mathcal{N}_b = \cup_{\mathcal{M}_b}\mathcal{N}_b(\mathcal{M}_b, 1/4)$, so that $|\mathcal{N}_b| \leq m_{L+1}^s$. We apply (44) to every $\widehat{a} \in \mathcal{N}_a$ and $\widehat{b} \in \mathcal{N}_b$ and use a union bound to get a constant $C_4 > 0$ such that with probability at least $1 - \delta$, for all $\widehat{a} \in \mathcal{N}_a, \widehat{b} \in \mathcal{N}_b$, and all $l$,

$$
\begin{aligned}
|\widehat{b}^\top W_{L+1}^\top \xi_l \widehat{a}| &\leq C_3\sqrt{\frac{1}{m_{L+1}}\log\frac{|\mathcal{N}_a| \cdot |\mathcal{N}_b| \cdot L}{\delta}} \\
&\leq C_3\sqrt{\frac{1}{m_{L+1}}\log\frac{m_{L+1}^s \cdot m_l^s \cdot L}{\delta}} \\
&= C_3\sqrt{\frac{1}{m_{L+1}}\left(s\log(m_{L+1}m_l) + \log\frac{L}{\delta}\right)} \\
&\leq C_4\sqrt{\frac{s}{m_{L+1}}\log m}. \qquad\qquad \left(s\log m = \Omega\left(\log\frac{L}{\delta}\right)\right)
\end{aligned}
$$

For arbitrary $a \in S^{m_l - 1}$ and $b \in S^{m_{L+1} - 1}$ with $\|a\|_0, \|b\|_0 \leq s$, there are $\widehat{a} \in \mathcal{N}_a$ and $\widehat{b} \in \mathcal{N}_b$ with $\|a - \widehat{a}\|_2, \|b - \widehat{b}\|_2 \leq 1/4$. Note that $g$ is linear in $a$ and $b$. Triangle inequality gives

$$
\begin{aligned}
|g_l(a, b)| &\leq |g_l(\widehat{a}, \widehat{b})| + |g_l(a, b) - g_l(\widehat{a}, \widehat{b})| \\
&\leq C_3\sqrt{\frac{s}{m_{L+1}}\log m_{L+1}} + |g_l(a, b) - g_l(\widehat{a}, b)| + |g_l(\widehat{a}, \widehat{b}) - g_l(\widehat{a}, b)| \tag{45}
\end{aligned}
$$

We have for any $\widehat{a}$,

$$
\begin{aligned}
|g_l(\widehat{a}, \widehat{b}) - g_l(\widehat{a}, b)| &= \left\|b - \widehat{b}\right\|_2 \left| g_l\left(\widehat{a}, \frac{b - \widehat{b}}{\left\|b - \widehat{b}\right\|_2}\right)\right| \\
&\leq \frac{1}{4}\sup_{\|b'\|_2 = \|a\|_2 = 1, \|a\|_0, \|b'\|_0 \leq s}|g_l(a, b')|. \tag{46}
\end{aligned}
$$

Similarly,

$$|g_l(a, b) - g_l(\widehat{a}, b)| \leq \frac{1}{4}\sup_{\|b\|_2 = \|a\|_2 = 1, \|a\|_0, \|b\|_0 \leq s}|g_l(a, b)|. \tag{47}$$

Taking supremum over the left hand side of (45) and using the bounds in (46) and (47) completes the proof. $\qquad\square$

### D.3 Proof of Lemma C.4

*Proof.* We notice that since $v = (1, \ldots, 1, -1, \ldots, -1)^\top$, we can write $g_l(a)$ as a sum of independent random variables in the following form:

$$g_l(a) = \sqrt{m_{L+1}} \sum_{j=1}^{m_{L+1}/2} \frac{1}{\sqrt{m_{L+1}}} \left[ \sigma(w_{L+1,j}^\top \xi_{l+1} a) - \sigma(w_{L+1,j+m_{L+1}/2}^\top \xi_{l+1} a) \right].$$

Since $\|\xi_{l+1} a\|_2$ is uniformly bounded by a constant, Hoeffding inequality yields a constant $C_3 > 0$ such that for fixed $a$, with probability at least $1 - \delta$, we have

$$g_l(a) \le C_3 \sqrt{m} \sqrt{\frac{1}{m} \log \frac{1}{\delta}}.$$

Let $\mathcal{M}$ be a fixed subspace of $\mathbb{R}^{m_l}$ with sparsity $s$, and let $\mathcal{N} = \cup_{\mathcal{M}} \mathcal{N}(\mathcal{M}, 1/2)$ be the union of all $1/2$-nets covering each $\mathcal{M}$ so that $|\mathcal{N}| \le m_l^s$. Using a union bound over all $\widehat{a} \in \mathcal{N}$ and $l$, we get that with probability at least $1 - \delta$, for all $\widehat{a} \in \mathcal{N}$ and all $l \le L$,

$$g_l(\widehat{a}) \le C_3 \sqrt{m} \cdot \sqrt{\frac{1}{m} \log \frac{|\mathcal{N}| \cdot L}{\delta}} \le C_5 \sqrt{s \log m}.$$

For arbitrary $a \in S^{m_l - 1}$ satisfying $\|a\|_0 \le s$, there is $\widehat{a} \in \mathcal{N}$ with $\|a - \widehat{a}\|_2 \le 1/2$. Since $g$ is linear,

$$|g_l(a)| \le |g_l(\widehat{a})| + |g_l(a - \widehat{a})| \le C_5 \sqrt{s \log m} + |g_l(a - \widehat{a})|. \tag{48}$$

For the second term, we have

$$|g_l(a - \widehat{a})| = \|a - \widehat{a}\|_2 \left| g_l \left( \frac{a - \widehat{a}}{\|a - \widehat{a}\|_2} \right) \right| \le \frac{1}{2} \sup_{\|a\|_2 = 1, \, \|a\|_0 \le s} |g_l(a)|.$$

Substituting this into (48) and taking supremums completes the proof. $\qquad\square$