[Reviews · NeurIPS 2019]

Reviewer 1



It appears to me that this paper is closely related to the paper by Zhang et al. [24]. The settings differs slightly: a slightly different type of residual networks and a different loss function are considered in [24]. Due to such differences, there are new issues that the current paper needs to deal with. However, the analysis basically follows the same outline and the high-level ideas are very similar. Therefore, the optimization result of this paper seems somewhat incremental based on that of [24]. The generalization result seems new to me, but it does not look surprising as it is based on a standard approach via Rademacher complexity. The results seem technically sound, as the proofs all look reasonable, although I did not check all of them carefully. The paper is well written in general, and the problem is important. * After author response: I have read the response, but it does not change my evaluation. I still feel that the contribution of this paper given the results of Zhang et al. [24] is not substantial enough, as changing the loss does not seem to make a huge difference and the generalization result given the optimization result does not seem to require very novel idea.

Reviewer 2



Originality: The assumptions and the overall proof ideas is similar to a previous work [5], as mentioned several times by the authors. However, extending the analysis to residual networks is non-trivial and novel, as far as I know. Quality: The claims and the theory seems sound, however, I have not checked all proofs in details. Clarity: I enjoyed reading the paper for the most part. Overall presentation is clear; specifically, the related work section is well done. However, lines 13-14 of the abstract requires more clarifications. Is this statement precise? Is there a result showing that the super logarithmic dependence on depth is un-avoidable in deep fully connected neural networks? Significance: the results is interesting in particular because it shows a non-trivial improvement in the generalization error when skip-connections are used. However, it is not clear if this stems from a sub-optimal analysis for deep feed-forward networks. ######## I have read the author feedback and considered it in my final evaluation. Overall, I think this paper is qualified to be published in NeurIPS.

Reviewer 3



The paper describes the proof for characterizing the fully connected deep ReLU residual network's optimization and generalization properties, trained with gradient descent following Gaussian initialization.Building on the work from [5] the authors demonstrate that over-parameterization forces gradient descent-trained networks to stay in a small neighborhood of initialization where the learned networks are guaranteed to find small surrogate training error, and come from a sufficiently small hypothesis class to guarantee a small generalization gap between the training and test errors. This allows for derivation of test error guarantees. The proof's show that provided we have sufficient overparameterization, gradient descent is guaranteed to find networks that have arbitrarily high classification accuracy. In comparison with the results of Cao and Gu [5] , the width m , number of samples n , step size η , and number of iterates K required for the guarantees for residual networks given in Theorem 3.5 and Corollary 3.7 all have (at most) logarithmic dependence on L. Additionally, the step size and number of iterations required for our guarantees are independent of the depth. Finally, the presence of skip connections in the network architecture removes the complications relating to the depth that traditionally arise in the analysis of non-residual architectures, and provides an explanation for why residual networks can be trained easily at all depths, unlike non-residual ones.

[Author Response · NeurIPS 2019]

We thank the reviewers for their clear and valuable comments on our work.

**Reviewer 1**

Question: Similarity to Zhang et al. [24] and explanation for loss guarantee at some iterate $k \in [K]$.

Response: As the reviewer notes, Zhang et al. focus on optimization and not generalization, and they study the squared
loss for regression while we examine cross entropy for classification. Extending an optimization result to generalization
requires a careful analysis that balances the training loss and the generalization gap, and therefore must depend on
the labels of the data. Zhang et al.'s analysis applies equally to data with labels randomly assigned, and thus a simple
Rademacher complexity argument based on their analysis cannot provide a meaningful generalization result.

The explanation for why we can only derive the guarantee for the loss at some iterate of the trajectory comes from the
difference in loss functions considered. A special property of the squared loss is that its derivative can be related to
the loss itself in a direct way: for a given sample $(x_i, y_i)$, the derivative of the loss for a prediction results in a term of
the form $(f_W(x_i) - y_i) \cdot \nabla f_W(x_i)$, where $f_W(\cdot)$ denotes the neural network output. By applying Jensen inequality,
this allows for the change in the empirical loss over a gradient descent step to be directly related to a function of the
empirical loss itself, while the cross-entropy loss analysis requires working with a surrogate loss defined in terms of its
derivative. This simplification is key to the analysis of the squared loss present in e.g. [1, 7] in addition to [24], and
allows for convergence at a linear rate and a straightforward formula for the empirical loss at a given iteration. For the
cross entropy, it is possible to show that the empirical loss is monotone decreasing (e.g. using line 287), but because the
derivative of the cross entropy loss is not as simply related to the loss itself, we are only able to get a guarantee that the
surrogate loss is sufficiently small at some point in the gradient descent trajectory, rather than its last iterate, by using
the telescoping argument described in lines 288–292. In short, there are some significant technical differences in the
analysis of optimization under the squared loss and generalization under cross entropy.

Question: Removal of logarithmic dependence on depth.

Response: We would like to note that Zhang et al.'s optimization result is not entirely independent of the depth $L$. They
require that $m \geq \max(L, \Omega(n^{24}\delta^{-8}d\log^2 m))$, and thus when $L \leq Cn^{24}\delta^{-8}d\log^2 m$ for some absolute constant $C$,
their result does not depend on $L$. We can derive a similar property if we assume $L \leq Cd\log m$.

**Reviewer 2**

Question: Fixing top layer weights and intuition for benefit for residual networks.

Response: In the revision, we will be clearer in explaining that our analysis can be extended to a trainable final layer
with a suitable random initialization, but that we chose to consider a fixed final layer for simplicity of exposition.
Additionally, we will be sure to emphasize that a key insight of our analysis is that the Lipschitz constant of deep
residual networks is independent of the depth, while all known analyses of fully connected networks have Lipschitz
constants growing at least polynomially in $L$, and that this is responsible for the simpler analysis and reduced depth
dependence in the residual architecture.

Question: Is super-logarithmic depth dependence necessary for fully connected networks?

Response: We are unaware of any results proving this necessity and we will be more careful to note this in the revision
of the paper.

Question: Context for Assumption 3.2 and Surrogate Loss.

Response: We will give additional context regarding these items in the revision of the paper.

**Reviewer 3**

Question: Comparison to the 'Generalization Bounds of SGD for Wide and Deep Neural Networks' paper.

Response: We thank the reviewer for pointing out the cited paper which recently appeared on arXiv. The 'Wide and
Deep' paper concerns optimization and generalization results for deep fully connected networks trained by stochastic
gradient descent, while ours concerns residual networks trained with gradient descent. The chief contribution of our
paper is a theoretically grounded explanation as to why deep residual networks are preferable to ones without residual
connections, and thus the consideration of a different architecture is a key component of our paper. From a technical
standpoint, the 'Wide and Deep' paper is based on a kernel/random feature method more similar to [3, 7, 8, 10]
rather than a direct trajectory analysis as in ours. The key generalization analysis in the 'Wide and Deep' paper is an
online-to-batch conversion that is specific to analyses of SGD, while ours is a uniform convergence argument for GD.
From the optimization perspective, our result is more similar to that of GD under smoothness assumptions rather than
SGD under Lipschitz and convexity assumptions. We will be sure to provide a comparison of our paper to the 'Wide
and Deep' paper in the revision.

[Meta-Review · NeurIPS 2019]

The paper derives generalization bounds for overparametrized deep residual networks learned by gradient descent from random initialization. All reviewers appreciate the importance of the topic of the paper. However, R1 and R3 feel that the contribution is too close to prior art, including [5] [24] and another NeurIPS submission. On the other hand, R2 thinks that the contributions relative to prior art are meaningful and vouches for acceptance. The rebuttal successfully addresses the differences: [24] focuses on optimization with squared loss, while this submission focuses on generalization with cross entropy loss. The Wide and Deep paper focuses on optimization and generalization for fully connected networks trained by SGD, while this paper focuses on residual networks trained with gradient descent. This AC sides with R2 assessment that there are enough differences relative to prior art to justify acceptance.